# Natural haplotypes of *FLM* non-coding sequences fine-tune flowering time in ambient spring temperatures in Arabidopsis

Ulrich Lutz[1], Thomas Nussbaumer[2†], Manuel Spannagl[2], Julia Diener[1], Klaus FX Mayer[2], Claus Schwechheimer[1*]

[1]Plant Systems Biology, Technische Universität München, Freising, Germany; [2]Plant Genome and Systems Biology, Helmholtz Zentrum München, German Research Center for Environmental Health, Neuherberg, Germany

**Abstract** Cool ambient temperatures are major cues determining flowering time in spring. The mechanisms promoting or delaying flowering in response to ambient temperature changes are only beginning to be understood. In *Arabidopsis thaliana*, *FLOWERING LOCUS M* (*FLM*) regulates flowering in the ambient temperature range and *FLM* is transcribed and alternatively spliced in a temperature-dependent manner. We identify polymorphic promoter and intronic sequences required for *FLM* expression and splicing. In transgenic experiments covering 69% of the available sequence variation in two distinct sites, we show that variation in the abundance of the *FLM-ß* splice form strictly correlate ($R^2$ = 0.94) with flowering time over an extended vegetative period. The *FLM* polymorphisms lead to changes in *FLM* expression (PRO2+) but may also affect *FLM* intron 1 splicing (INT6+). This information could serve to buffer the anticipated negative effects on agricultural systems and flowering that may occur during climate change.

**\*For correspondence:** claus.
schwechheimer@wzw.tum.de

**Present address:**
[†]Computational Systems Biology, University of Vienna, Vienna, Austria

**Competing interests:** The authors declare that no competing interests exist.

## Introduction

Plants are sessile organisms that have adapted to their habitats to optimize flowering time and thereby guarantee reproductive success and survival. Temperature is one major cue controlling flowering time, particularly before the winter but also during spring when ambient cool temperatures generally delay and warm temperatures promote flowering. Different molecular pathways controlling flowering in different temperature ranges and environments have been genetically dissected (*Capovilla et al., 2015*; *Verhage et al., 2014*).

In many plant species and in many accessions of the plant model species *Arabidopsis thaliana* (Arabidopsis), the well-studied vernalization pathway prevents premature flowering before the long cold periods of the winter (*Song et al., 2012*). In Arabidopsis, flowering of winter-annual accessions without vernalization is strongly delayed by the MADS-box transcription factor *FLOWERING LOCUS C* (*FLC*) (*Song et al., 2012*; *Michaels and Amasino, 1999*, *2001*; *Johanson et al., 2000*). FLC forms a repressor complex through interactions with the MADS-box transcription factor SVP (SHORT VEGETATIVE PHASE) to repress the transcription of the flowering promoting genes *FLOWERING LOCUS T* (*FT*) and *SUPPRESSOR OF OVEREXPRESSION OF CO1* (*SOC1*) (*Li et al., 2008*; *Lee et al., 2007*). As a result of the prolonged exposure to cold temperatures during winter, *FLC* abundance is gradually reduced, predominantly through epigenetic mechanisms, and flowering repression is gradually relieved (*Song et al., 2012*). Substantial natural variation in the vernalization-dependent expression

of *FLC* has already been described and characterized in many Arabidopsis accessions (*Coustham et al., 2012*; *Li et al., 2014*).

During spring, ambient temperature is an important climatic factor. Temperature changes by only a few degree Celsius (°C) during cold or warm spring periods shift flowering time in many plant species. Since temperature changes associated with global warming could lead to similar flowering changes and threaten agricultural production systems, elucidating the ambient temperature pathway has recently received increased attention (*Moore and Lobell, 2015*; *Wheeler and von Braun, 2013*; *Jagadish et al., 2016*).

The complexities of the ambient temperature flowering pathway are just beginning to be understood (*Capovilla et al., 2015*; *Verhage et al., 2014*). After vernalization or in vernalization-insensitive accessions, flowering in ambient temperatures is largely under control of the *FLC*-related *FLOWERING LOCUS M* (*FLM*) and residual *FLC* retains only a minor role (*Gu et al., 2013*; *Balasubramanian et al., 2006*; *Blázquez et al., 2003*; *Lee et al., 2013*). *FLM* controls flowering in the range between 5°C and 23°C and FLM can form, just like FLC, a flowering repressive complex with SVP (*Lee et al., 2013*; *Posé et al., 2013*). Besides *FLM,* as the dominant regulator, ambient temperature flowering is also regulated by the *FLM*-homologs *MAF2 - MAF4* (*Li et al., 2008*; *Gu et al., 2013*; *Lee et al., 2013*; *Ratcliffe et al., 2003*; *Airoldi et al., 2015*). Conversely, loss-of-function mutations of *SVP* lead to early and temperature-insensitive flowering in the range between 5°C and 27°C (*Lee et al., 2007*, *2013*).

Within the ambient temperature range, *FLM* is differentially spliced in a temperature-dependent manner through the alternative use of the exons 2 (*FLM-ß*) and 3 (*FLM-δ*). Based on transgenic experiments, a model was proposed according to which FLM-ß and FLM-δ function antagonistically with SVP. Accordingly, FLM-ß engages in flowering repressive interactions with SVP in cooler temperatures. Conversely, FLM-δ would outcompete FLM-ß in warmer temperatures and form heterodimers with SVP unable to bind DNA and repress flowering (*Lee et al., 2013*; *Posé et al., 2013*). Recent data suggest that this attractive model, based solely on transgenic experiments, may not be valid in natural contexts (*Sureshkumar et al., 2016*; *Lutz et al., 2015*).

As yet, it has remained largely unknown whether genetic variation of the *FLM* gene locus plays a role in flowering time regulation across Arabidopsis accessions and if so, which variations determine basal and temperature-dependent *FLM* expression and splicing. Although two *FLM* deletion alleles conferring temperature-insensitive early flowering were identified in the Arabidopsis accessions Niederzenz-1 (Nd-1) and Eifel-6 (Ei-6), their limited demographic and genetic spread indicated that *FLM* deletions may be disadvantageous (*Werner et al., 2005*; *Balasubramanian and Weigel, 2006*). A first *FLM* expression variant was determined from the early flowering accession Killian-0 (Kil-0). In Kil-0, a LINE retrotransposon insertion in *FLM* intron 1 caused premature transcription termination, aberrant *FLM* splicing, consequently reduced *FLM* expression and earlier flowering. This phenotype was especially prominent at a temperature of 15°C , which is closer to the average temperature in the native range of the species than the commonly used 21°C (*Lutz et al., 2015*; *Hoffmann, 2002*; *Weigel, 2012*). The subsequent identification of nine further accessions with an identical LINE insertion was suggestive for a recent adaptive selective sweep (*Lutz et al., 2015*). It was thus concluded that *FLM* expression-modulating alleles are advantageous for flowering time adaptation in the ambient temperature range, particularly since there are no other pleiotropic growth changes observed in *FLM* mutant alleles.

Previous work had shown that an intronless *FLM* cDNA expressed from a *FLM* promoter fragment was unable to rescue the *flm* mutant phenotype. Since this suggested that important information for *FLM* expression may reside in *FLM* non-coding sequences, we examined non-coding sequence polymorphisms with a potential role in controlling *FLM* expression and splicing. Using phylogenetic footprinting, we found conserved *FLM* promoter and intron 1 regions essential for *FLM* expression. By association analysis using polymorphism data from ≈800 Arabidopsis accessions, we identified a small polymorphic region in the *FLM* promoter and a highly polymorphic nucleotide triplet in *FLM* intron 6 controlling basal and temperature-dependent *FLM* expression. Small changes in the relative abundance of the *FLM-ß* splice variant dynamically modulated flowering at 15°C over a range of 15 leaves. When tested in a homogenous genetic background, *FLM* abundance correlated almost perfectly with flowering time ($R^2 = 0.94$) and contributed ($R^2 = 0.21$) to flowering time variation in heterogeneous natural Arabidopsis populations. Our data suggest that *FLM-ß* is an important determinant of flowering during cool or warm spring periods.

# Results

## Intronic sequences are required for basal and temperature-sensitive *FLM* expression

Temperature-dependent changes in *FLM* abundance and alternative splicing are critical for flowering time control in ambient temperatures in Arabidopsis. How *FLM* expression is regulated and whether Arabidopsis accessions have employed differential *FLM* expression and splicing to adapt to different temperature climates remained to be shown. We previously found that Arabidopsis transgenic lines expressing an intronless *FLM* from a functional *FLM* promoter fragment cannot express *FLM* to detectable levels and fail to rescue the flowering time phenotype of a *flm-3* loss-of-function mutant (*Lutz et al., 2015*). We subsequently compared *FLM* expression in transgenic lines expressing a genomic *FLM* fragment, obtained from the Columbia-0 (Col-0) wild type, containing all six introns (pFLM::gFLM; FLM^Col-0^) with lines expressing the *FLM-ß* (pFLM::FLM-ß) or *FLM-δ* (pFLM::FLM-δ) splice variants retaining intron 1 but lacking all other introns (*Figure 1A*). The presence of intron 1 was sufficient to restore the basal expression of *FLM* at the ambient temperatures 15°C, 21°C, and 27°C, but introns 2–6 were required for the temperature-sensitive regulation of the splice variants *FLM-ß* and *FLM-δ* (*Figure 1B*). We concluded that intron 1 may contain critical information for *FLM* expression and introns 2–6 may contribute to temperature-dependent *FLM* expression.

## Phylogenetic footprinting pinpoints essential regions for basal expression in the promoter and the first intron

To identify non-coding regions important for *FLM* expression, we performed multiple sequence alignments of Arabidopsis *FLM* with its closest sequence homologue *MAF3* and *FLM* homologues from five other *Brassicaceae* species (*Figure 2—figure supplement 1A,B*) (*Martinez-Castilla and Alvarez-Buylla, 2003*; *Van de Velde et al., 2014*). Within the non-coding sequences, we detected

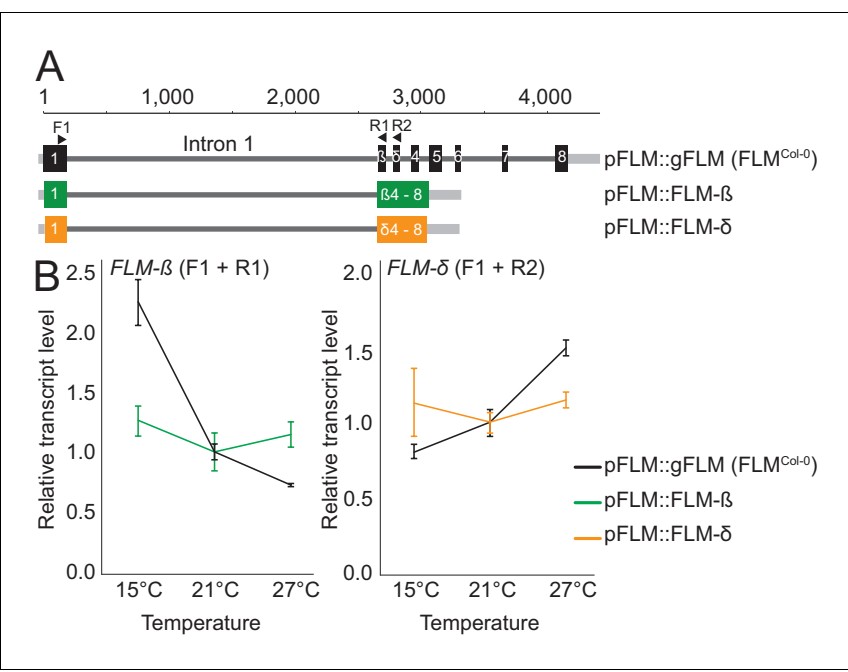

**Figure 1.** *FLM* intronic sequences determine basal and temperature-dependent *FLM* expression. (**A**) Schematic representation of the pFLM::gFLM (FLM^Col-0^), pFLM::FLM-ß, and pFLM::FLM-δ constructs. Scale is set to 1 according to the A of the ATG start codon. Black boxes represent exons, grey lines 5'- or 3'-untranslated regions, dark-grey lines introns. Arrows indicate primer binding sites; the forward primer F1 was used to amplify *FLM-ß* and *FLM-δ* with reverse primers R1 and R2, respectively. (**B**) Mean and SD (three biological replicates) of qRT-PCR analyses of *FLM-ß* and *FLM-δ* expression in ten day-old plants harbouring the transgenes shown in (**A**). For normalization, the respective values measured at 21°C were set to 1.

one promoter region of 250 bp and two intron 1 regions of 373 and 101 bp with increased sequence conservation (>60%) (*Figure 2A* and *Figure 2—figure supplement 1*). We then generated PRO$^{\Delta 225bp}$, INT1$^{\Delta 373bp}$, and INT1$^{\Delta 101bp}$ transgenic lines expressing pFLM::gFLM variants (FLM$^{Col-0}$) with deletions of the three regions in the *FLM* deletion accession Nd-1 to measure the effects on *FLM-ß* and *FLM-δ* expression at 15°C and 23°C (*Figure 2B* and *Figure 2—figure supplement 2*) (*Werner et al., 2005*). To normalize for variability between the transgenic lines, we examined pools of independent T2 segregating lines (n = 21–34). We validated this pooling strategy by demonstrating that the established behaviour of *FLM* in the Col-0 and Kil-0 accessions could be faithfully recapitulated when performing equivalent analyses with T2 lines expressing gFLM$^{Col-0}$ and gFLM$^{Col-0}$ bearing the Kil-0 LINE insertion (*Lutz et al., 2015*) (*Figure 2—figure supplement 2C,D*). While we detected a strong downregulation of *FLM* at 15°C and 23°C in PRO$^{\Delta 225bp}$ and INT1$^{\Delta 373bp}$ lines, the deletion in INT1$^{\Delta 101bp}$ had comparably minor effects on *FLM* expression (*Figure 2C,D*). This indicated that the proximal 225 bp promoter and the 337 bp intron 1 regions contained important sequences for basal *FLM* expression. Subsequent experiments showed that the 254 bp promoter fragment alone was, however, not sufficient to confer *FLM* expression of a genomic *FLM* fragment (PRO$^{254bp}$; *Figure 2B–D*).

## Non-coding sequence variation of major *FLM* haplotypes influences *FLM* expression

To find further non-coding determinants for *FLM* expression, we analysed *FLM* nucleotide variation in 776 sequenced Arabidopsis accessions (*The 1001 Genomes Consortium, 2016*). We identified 45 promoter and intronic SNPs with a minor allele frequency (MAF)≥5% and used these to define ten major haplotypes (H1–-H10) representing 379 (49%) accessions (*Figure 3—figure supplement 1A,B*; *Supplementary file 1*). We defined an initial set with 41 accessions by selecting five to twelve accessions from six of the ten haplotype groups (*Figure 3A*). Since introns 2–6 seemed important for temperature-sensitive *FLM* expression, we added 11 accessions with varying intron 2–6 haplotypes (HI2-6) but identical intron 1 haplotype (HI1; *Figure 3A* and *Figure 3—figure supplement 1D–G*). Finally, we added Col-0 (H3) and Kil-0 (H1) due to their well characterized *FLM* regulation to obtain a representative *FLM* haplotype set with ultimately 54 accessions (*Supplementary file 2*). We assured, by analytical PCR, that none of these accessions, except Kil-0, carried the previously described intron 1 LINE insertion (*Lutz et al., 2015*).

To find polymorphisms regulating *FLM* expression and splicing, we performed a genotype-phenotype association analysis. Since the vernalization pathway strongly delayed flowering in 24 of the 54 accessions and consequently suppressed *FLM* effects, we used *FLM* expression as a phenotype for the association analysis (*Figure 3—figure supplement 2A*). To ascertain that *FLM* expression was not affected by *FLC*, we examined *FLM* in non-vernalized wild type Col-0 and *flc-3* mutants as well as in Col-0 carrying a functional vernalization module (*Michaels and Amasino, 1999*). Concurrent with previous reports, we did not detect an influence of *FLC* on *FLM* transcript abundance in our conditions (*Figure 3—figure supplement 2B*) (*Scortecci et al., 2001*; *Ratcliffe et al., 2001*). We then obtained *FLM* expression data from the *FLM* haplotype set and measured total *FLM* transcript levels as well as *FLM-ß* and *FLM-δ* levels at 15°C or 23°C (*Supplementary file 3*). In line with the reported behaviour of *FLM* in Col-0, we observed that *FLM-ß* expression decreased (on average 0.6 fold) and *FLM-δ* increased (on average 1.4 fold) with increasing temperature in most accessions (*Figure 3B* and *Supplementary file 3*) (*Lee et al., 2013*; *Posé et al., 2013*; *Lutz et al., 2015*). At the same time, we also observed substantial variation in *FLM-ß* and *FLM-δ* expression between the accessions of the *FLM* haplotype set, inviting the conclusion that non-coding sequence variation modulates *FLM* expression (*Figure 3B* and *Supplementary file 3*). Since total *FLM* transcript levels strongly correlated with *FLM-ß* but not with *FLM-δ* abundance at 15°C and at 23°C, it could be suggested that *FLM-ß* represents the major *FLM* form among the *FLM* transcripts (*Figure 3—figure supplement 3A,B*).

## Association analysis identifies polymorphic sites with potential *FLM* regulatory functions

Using the *FLM* haplotype set, we next performed association tests between *FLM* expression at 15°C and 23°C or expression ratios derived from these values and an extended set of 119 polymorphisms

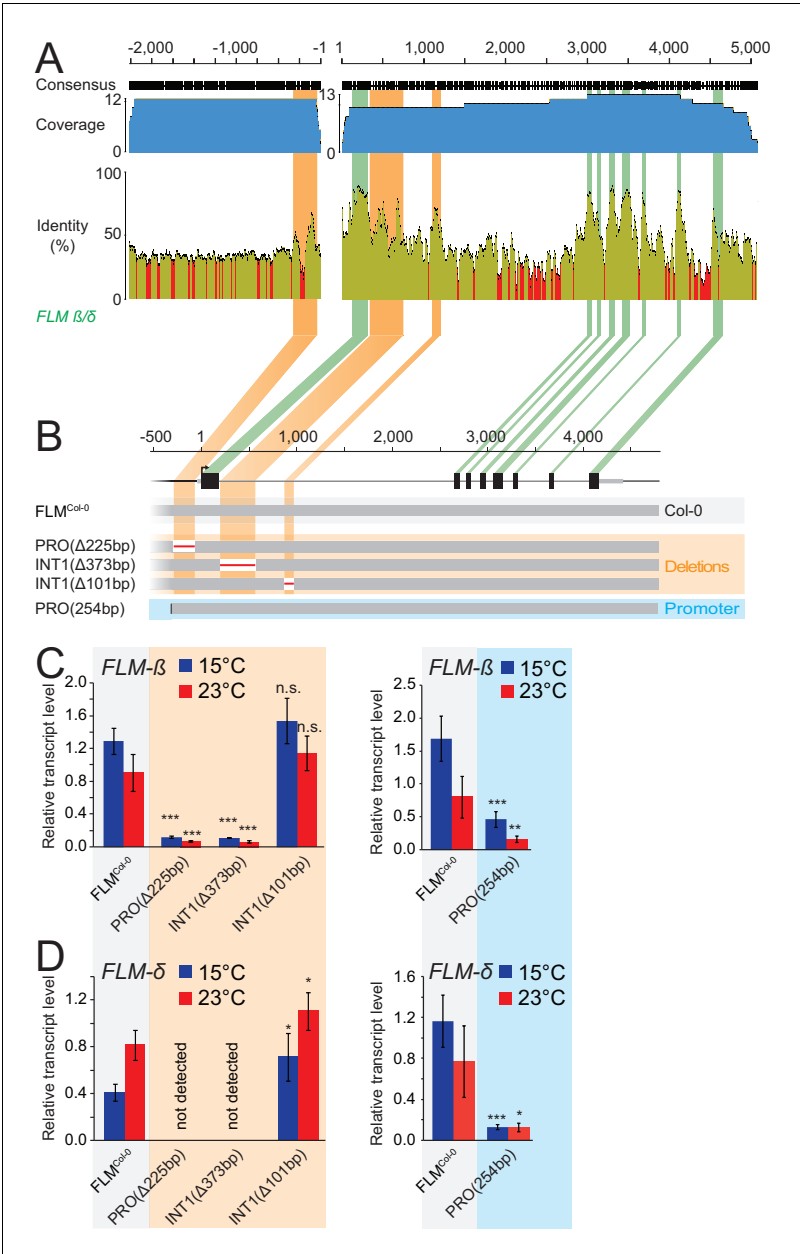

**Figure 2.** Phylogenetic footprinting identifies promoter and intron 1 regions required for *FLM* expression. (A) Phylogenetic footprinting of promoter and genomic regions of *FLM* and putative *FLM* orthologs from six *Brassicaceae* species. Coverage is shown in blue, identities are shown in ocre (≥30%) and red (<30%). Exons are displayed in green, regions with high non-coding sequence conservation are displayed in orange. (B) Schematic illustration of the *FLM* genomic region and FLM<sup>Col-0</sup> transgenic variants used for expression analysis. (C) and (D) Mean and SD (four replicate pools with five to ten independent T2 transgenic lines) of qRT-PCR analysis of *FLM-β* (C) and *FLM-δ* (D) expression at 15°C and 23°C in ten day-old seedlings of 21–40 bulked T2 transgenic lines. Student's t-tests: *, p≤0.05; **, p≤0.01; ***, p≤0.001; n.s., not significant.

The following figure supplements are available for figure 2:

**Figure supplement 1.** Phylogenetic footprinting of promoter and genomic regions of *FLM* and putative *FLM* orthologs from six *Brassicaceae* species.

**Figure supplement 2.** The gFLM<sup>Col-0</sup> and gFLM<sup>Col-0</sup>+LINE transgenes faithfully recapitulate *FLM* expression in the Col-0 and Kil-0 accessions.

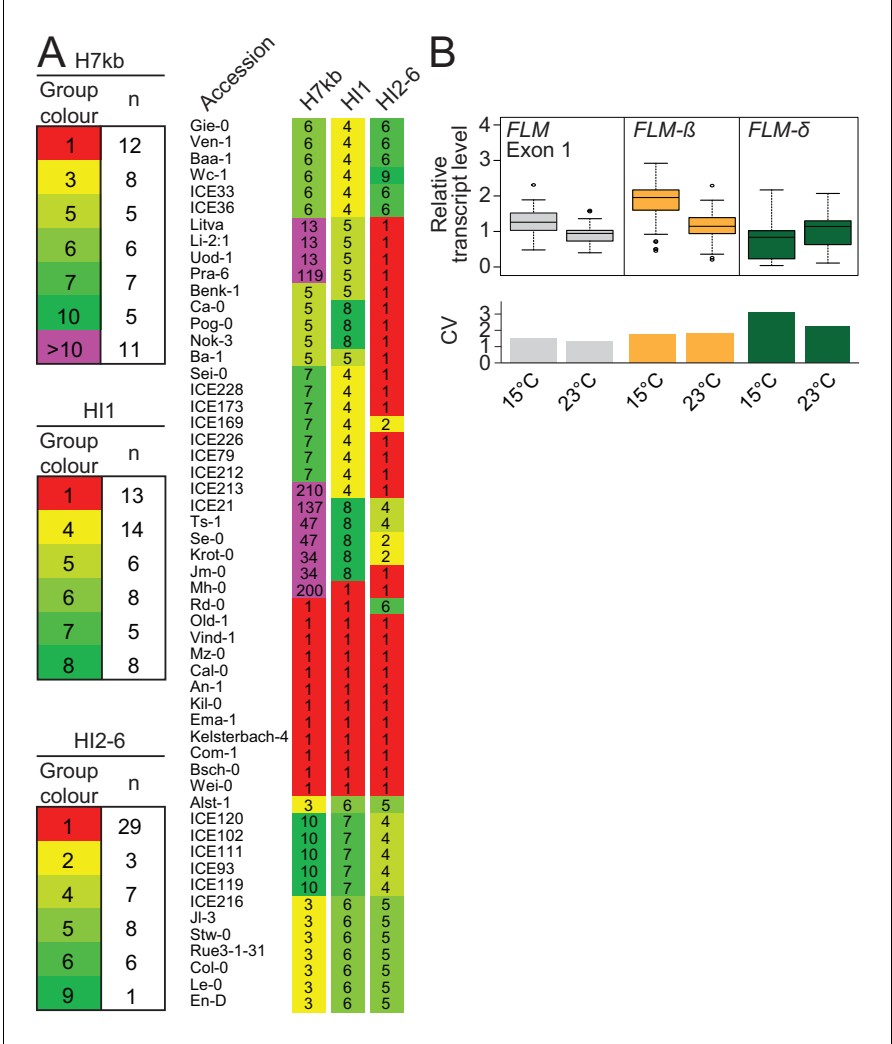

**Figure 3.** *FLM* haplotype analysis of 776 accessions identifies major haplotypes determined by non-coding variation. (**A**) Haplotype group affiliation of 54 accessions of the *FLM* haplotype set based on 45 SNPs for either the 7 kb *FLM* locus (H7kb), intron 1 (HI1) or introns 2–6 (HI2-6). Group numbering and colouring are according to group size. (**B**) Summarized expression values of total *FLM* (exon 1), *FLM-ß*, and *FLM-δ* expression of the *FLM* haplotype set. Outliers were determined based on 1.5 x IQR (interquartile range). The coefficient of variation (CV) is shown in the lower graph.

The following figure supplements are available for figure 3:

**Figure supplement 1.** Haplotype analysis based on 45 SNPs from 776 accessions.

**Figure supplement 2.** Flowering time of the accessions of the *FLM* haplotype set suggest a vernalization requirement for many accessions.

**Figure supplement 3.** Correlation analysis identifies *FLM-β* as the major *FLM* transcript at 15°C and 23°C in the *FLM* haplotype set accessions.

**Figure supplement 4.** 119 polymorphic sites among the accessions of the *FLM* haplotypes set.

**Figure supplement 5.** Results from the simple single locus association test between the 119 polymorphic sites with *FLM* expression values.

**Figure supplement 6.** Linkage analysis for the simple single locus association test.

*Figure 3 continued on next page*

*Figure 3 continued*

**Figure supplement 7.** Effects of the PRO1, PRO2 and INT6 polymorphisms on *FLM* gene expression.

along the 7 kb *FLM* region, which included the 45 SNPs, but also low frequent SNPs (MAF>1%) and indels (≤2 bp; MAF>1%) (*Figure 3—figure supplement 4*). The association analysis detected polymorphisms with significant associations (p<0.001) (*Figure 3—figure supplement 5* and *Supplementary file 4*). We decided to investigate the potential role of a single base pair deletion (PRO1$^{T/-}$; bp −215) and a genetically slightly linked SNP (PRO2$^{A/C}$; bp −93), because they were both located in the proximal part of the *FLM* promoter. Further, we investigated three genetically unlinked nucleotides because they were positioned as a highly diverse nucleotide triplet in intron 6 (INT6$^{A/C-A/C-A/T/C}$; bp +3975–+ 3977) in an otherwise conserved sequence context (*Figure 4A*, *Figure 3—figure supplements 5–7*, *Supplementary file 5*).

Strikingly, the PRO2 and INT6 sites were directly flanked or in close proximity to a total of four PolyA motifs of variable length ([A]$_{7-11}$), one of which represented a so-called CArG-box, a potential binding site for MADS-box transcription factors (*Zhang et al., 2016*). Two more PolyA motifs resided in introns 3 and 5 (*Figure 4D* and *Figure 4—figure supplement 1A*). Since PolyA motifs had been reported to be important for gene expression, we reasoned that these motifs could be relevant for *FLM* regulation, in isolation or in combination with the highly significant non-coding variations (*O'Malley et al., 2016*; *Horton et al., 2012*).

To understand the sequence variation surrounding the PRO1, PRO2 and INT6 sites, we reanalysed available Arabidopsis genome sequences (*The 1001 Genomes Consortium, 2016*). Besides the PRO1 1 bp deletion, the region surrounding PRO1 was conserved and we therefore focussed on the PRO1$^{T/-}$ deletion polymorphism (*Figure 4A*). Few bases up- and downstream of PRO2 (bp −102 to bp −92), we detected four additional highly diverse SNPs and designated this region PRO2+. We also identified additional frequent haplotypes of the INT6 triplet (INT6+). Apart from the two PolyA motifs located at PRO2+ and INT6+, the four remaining PolyA motifs were conserved across all accessions examined (*Figure 4B*).

## *FLM* non-coding sequence variation regulates *FLM* abundance in a homogenous transgenic background

To examine the effects of the non-coding sequence variations in a homogenous background, we transformed the deletion accession Nd-1 with FLM$^{Col-0}$ variants carrying the PRO1, PRO2+ and INT6 + polymorphisms or deletions of PRO2+ (PRO2+$^{Δ16bp}$) and INT6+ (INT6+$^{Δ17bp}$) (*Figure 4C,D*). Additionally, we transformed variants with deletions of between two and six PolyA stretches (PolyA_2xΔa, PolyA_2xΔb, PolyA_3xΔ, PolyA_4xΔ, PolyA_5xΔ, PolyA_6xΔ; *Figure 4D* and *Figure 4—figure supplement 1A*). Using pooled T2 segregating lines (n = 18–45) for each of the transgenes, we analysed the effects of the sequence variation on *FLM-ß* and *FLM-δ* expression at 15°C and 23°C. The PRO1$^-$ deletion, as present in the FLM$^{Col-0}$ reference, did not reveal differences in *FLM* transcript levels when compared to PRO1$^T$ (FLM$^{Col-0}$) (Fig. *Figure 4—figure supplement 1B,C*). However, two PRO2 + (PRO2+$^{GGAAC}$, PRO2+$^{AAACC}$) and three INT6+ variants (INT6+$^{ACA}$, INT6+$^{CAC}$ and INT6+$^{AAA}$) displayed an upregulation of *FLM-ß* at 15°C when compared to FLM$^{Col-0}$ (*Figure 4E*). Increases and decreases of *FLM-δ* levels largely followed those of *FLM-ß* levels except for INT6+$^{AAA}$, which showed an upregulation exclusively of *FLM-ß* (*Figure 4E,F*). The deletion variant PRO2+$^{Δ16bp}$ showed upregulation of *FLM-ß* but INT6+$^{Δ17bp}$ did not have significantly altered *FLM-ß* and *FLM-δ* levels when compared to FLM$^{Col-0}$ (*Figure 4E,F*). We concluded that the identity of these regions rather than their presence controlled *FLM* expression and temperature-sensitive *FLM* regulation. The PolyA motif deletion variants showed gradually decreasing *FLM-ß* and *FLM-δ* levels with an increasing number of deletions while the deletion of all six PolyA motifs (PolyA_6xΔ) had an especially strong effect on *FLM-ß* expression and its temperature-sensitive expression (*Figure 4E,F* and *Figure 4—figure supplement 1B,C*). Thus, the PolyA motifs possess a regulatory role but may function in a redundant manner.

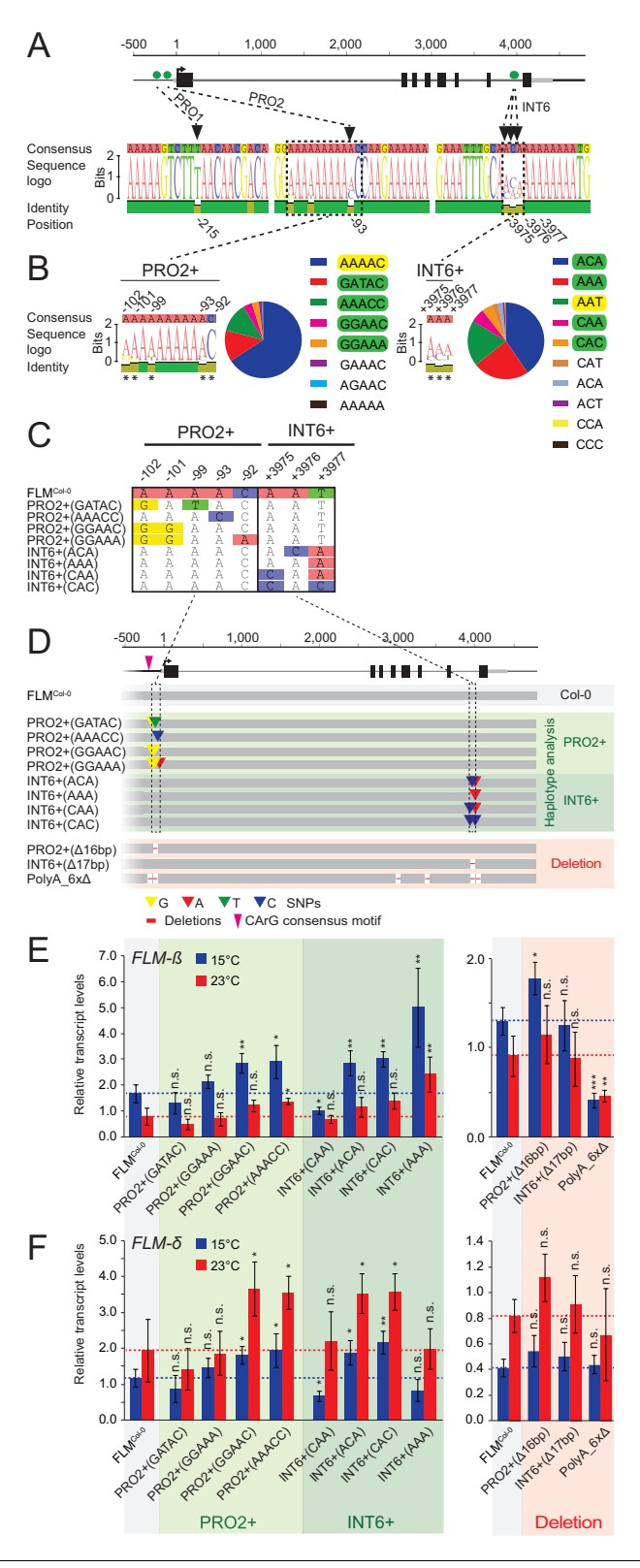

**Figure 4.** Polymorphisms in PRO2+ and INT6+ sites influence basal and temperature-dependent *FLM* expression. (**A**) Schematic representation of the *FLM* genomic locus as shown in *Figure 1*. Green dots indicate the positions of the PRO1, PRO2, and INT6 sites that were chosen for further investigation. Sequence logos display the allele frequencies at these sites among the 54 accessions of the *FLM* haplotype set. (**B**) Sequence logos and allele

*Figure 4 continued on next page*

*Figure 4 continued*

frequency distribution of PRO2+ and INT6+ polymorphisms among 840 Arabidopsis accessions. All polymorphic residues are marked with asterisks and the respective allele frequencies are indicated by the sequence logo. The Col-0 reference haplotype is marked in yellow, haplotypes chosen for further investigation are marked in green. (**C**) and (**D**) Alignment of the PRO2+ and INT6+ polymorphisms (**C**) and schematic representation of the FLM[Col-0] reference construct (pFLM::gFLM) as well as the FLM[Col-0] variants selected for transgenic analysis in the *FLM* deletion accession Nd-1. Bases that differ between FLM[Col-0] and the variants are coloured. Deletions in the deletion constructs of the PRO2+ and INT6+ sites as well as the PolyA motifs are displayed with red lines (not drawn to scale). (**E**) and (**F**) Mean and SD (four replicate pools with four to eleven independent T2 transgenic lines) of qRT-PCR analyses of *FLM-β* (**E**) and *FLM-δ* (**F**). For easier comparison, the values obtained with FLM[Col-0] are displayed as red and blue dotted lines. Student's t-tests: *p≤0.05; **p≤0.01; ***p≤0.001; n.s., not significant.

The following figure supplement is available for figure 4:

**Figure supplement 1.** Increasing the number of PolyA motif deletions results in the gradual reduction of *FLM-ß* expression.

## *FLM-ß* levels and flowering time strongly correlate in a linear manner

The phylogenetic footprinting and association analyses resulted in the identification of polymorphisms and regions that significantly altered *FLM* expression when present in a homogenous molecular and genomic context. To correlate *FLM* expression with flowering, we measured flowering time in T2 transgenic lines of eight variants with significantly different *FLM* abundance (*Figure 5A* and *Figure 5—figure supplement 1A–C*). We detected a very strong correlation between *FLM-ß* transcript levels and flowering time ($R^2 = 0.94$) in plants grown at 15°C. Flowering time responded in a linear manner to *FLM-ß* levels with 17 to 30 rosette leaves until flowering in the range of relative transcript levels from 0.6 to 2.4 (*Figure 5B*). Further, the low expression variants PolyA_6xΔ, INT1[Δ373bp], and PRO[Δ225bp] flowered as early as the deletion accession Nd-1 suggesting that there must be a critical lower threshold for *FLM-ß* to be effective (*Figure 5A,B*). The correlation was much lower for *FLM-δ* ($R^2 = 0.70$) suggesting that *FLM-ß* is the major determinant for flowering time in these conditions (*Figure 5—figure supplement 1D*). Within the range of lines examined, we did not detect a saturating effect at the upper expression level.

## The INT6+[CAA] polymorphism confers temperature-insensitive *FLM* expression and flowering

Introns 2–6 were required for temperature-sensitive *FLM* expression (*Figure 1*). When we statistically tested the interaction between genotype and temperature with a multiple linear model, we found that temperature-sensitive *FLM-ß* regulation was significantly reduced in INT6+[CAA] between 15°C and 23°C and when compared to the FLM[Col-0] control variant (p=0.012) (*Figures 4E* and *6A,B*). In line with the prediction, flowering of this variant was indeed less sensitive to temperature changes when tested at 15°C and 23°C in homozygous T3 progeny plants and compared to the FLM[Col-0] reference (*Figure 6C,D*). Thus, temperature-independent *FLM-ß* expression changes correlate with temperature-insensitive flowering in the selected temperature range.

## Differential abundance of *FLM* splice forms can correlate with changes in *FLM* transcription or alternative splicing

To understand whether the same or different molecular mechanisms are the basis of altered *FLM* expression in *FLM* variants, we estimated *FLM* transcription of variants with strongly altered abundance of processed *FLM* abundance by measuring levels of unprocessed *FLM* pre-mRNA from plants grown at 15°C. When compared to the levels of processed *FLM* mRNA, the PRO[Δ225bp] and INT1[Δ373bp] variants showed similarly strong reductions of unprocessed pre-mRNA, suggesting that the respective deletion polymorphisms directly affect *FLM* transcription (*Figures 4E,F and 7A*). In turn, the INT6+[CAA] and PolyA_6xΔ lines had reduced *FLM* mRNA levels but, when compared to FLM[Col-0], did not show substantial changes in unprocessed pre-mRNA levels (*Figures 4E,F and 7B*). Since this indicated that post-transcriptional events may be affected in these variants, we tested for the abundance of differential polyadenylated splice variants after semi-quantitative 3'-

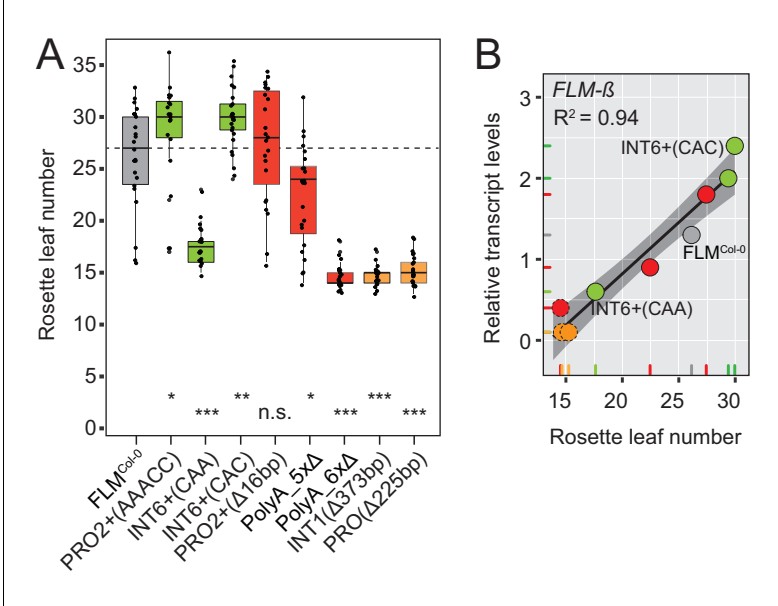

**Figure 5.** *FLM-ß* expression shows high correlation with flowering time of transgenic plants. (**A**) Box plot of quantitative flowering time analysis (rosette leaf number) of independent T2 transgenic lines at 15°C and long day photoperiod. Ten plants of three replicate pools were analysed for each construct. The data were corrected for the anticipated 25% of non-transgenic Nd-1 segregants (see *Figure 5—figure supplement 1D* for the uncorrected analysis). Single values are shown as jittered dots, the colour represents the type of variant as introduced in *Figure 4D*, the median of the FLM[Col-0] reference is indicated as dotted line. Wilcoxon rank test: *p≤0.05; **p≤0.01; ***p≤0.001; n.s., not significant. (**B**) Correlation (simple linear regression) between *FLM-ß* and qRT-PCR expression data as presented in *Figure 5—figure supplement 1A,B* and flowering time analysis as shown in (**A**). Datapoints of INT6+ variants with contrasting *FLM* expression and FLM[Col-0] are designated. The color code corresponds to (**A**). The variants PolyA_6x$\Delta$, INT1[$\Delta$373bp], and PRO[$\Delta$225bp], which do not respond in a linear manner are shown as dotted circles. The shaded areas indicate the 95% confidence intervals; p<0.0001. Note, that the flowering time data shown in (**A**) were corrected by removing values of non-transgenic T2 segregants. An analysis using the uncorrected data is shown in *Figure 5—figure supplement 1E,F*.

The following figure supplement is available for figure 5:

**Figure supplement 1.** *FLM-ß* expression correlates with flowering time in transgenic lines carrying different *FLM* alleles.

RACE-PCR and sequencing of the cloned PCR products (*Figure 7—figure supplement 1A*). There, we detected a relative reduction of *FLM-ß* transcripts in INT6+[CAA] and PolyA_6x$\Delta$ that was accompanied by increases in the abundance of two polyadenylated transcripts containing exon 1 and intron 1 (E1I1p) that had already been noted in an earlier publication (*Figure 7C,D*) (*Lutz et al., 2015*). Importantly, we did not identify a single *FLM-δ* clone among the 163 sequenced cDNAs. The relative increase in E1I1p transcripts could also be independently confirmed by E1I1p-specific qRT-PCRs and suggested in summary that splicing site choice at the exon 1 - intron 1 junction is changed in the INT6+[CAA] and PolyA_6x$\Delta$ alleles (*Figure 7—figure supplement 1B,C*). In relation to all exon 1-containing transcripts, the overall abundance of these intron 1-containing transcripts was comparatively low (*Figure 7—figure supplement 1C*).

## PRO2+ and INT6+ polymorphisms contribute to global variation of *FLM* levels

The nine PRO2+ and INT6+ haplotypes tested in transgenic experiments were present in 579 (69%) of all 840 accession with available genome sequence information (*Figure 8—figure supplement 1A*). To examine whether these haplotypes explain natural variation of *FLM-ß* levels in natural accessions, we randomly selected an experimental population of 94 accessions (2 to 14 accessions per

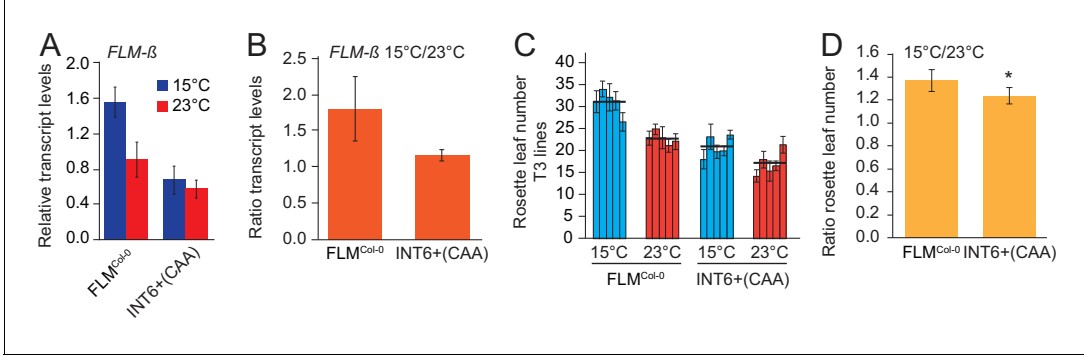

**Figure 6.** The INT6+[CAA] polymorphism reduces temperature-sensitivity of *FLM* expression and flowering. (A) qRT-PCR analysis and (B) expression ratios of *FLM-ß* of FLM[Col-0] (INT6+[AAT]) and INT6+[CAA] grown in 15°C and 23°C. Statistical tests of temperature-sensitivity are described in the text. (C) Means and SD of quantitative flowering time analysis (rosette leaf number). n = 5 (15°C) and 8 (23°C) replicates from five independent homozygous T3 transgenic lines grown at 15°C and 23°C in long day photoperiod. (D) Mean and SD of ratios of rosette leaf numbers from the analysis shown in (C). Student's t-test: *p<0.05; **p≤0.01.

PRO2+/INT6+ group, average 10) with a broad genetic and geographic distribution (*Figure 8—figure supplement 1B,C*). We next determined *FLM-ß* expression and, following quality filtering, grouped 85 accessions according to their PRO2+ or INT6+ haplotypes (*Supplementary file 6A*).

We found that the PRO2+/INT6+ haplotype significantly affected *FLM-ß* transcript levels (*Figure 8—figure supplement 1D*). When we integrated the average values from these natural accessions with the respective values from the transgenic analysis, we detected a positive, however not significant correlation, a likely consequence of the small number of datapoints (*FLM-ß*, $R^2 = 0.13$) (*Figure 8—figure supplement 1E*).

To examine the correlation between *FLM-ß* expression and flowering time, we determined flowering time of accessions at 15°C. To avoid strong interference from the vernalization pathway, we selected 27 genetically diverse summer-annual accessions, which initiate flowering without the need of vernalization (*Figure 8—figure supplement 2A*, *Supplementary file 6B*). We measured *FLM-ß* transcript levels and flowering time at 15°C. As residual *FLC* transcript in these summer-annual accessions may still affect flowering time, we also determined *FLC* transcript levels (*Coustham et al., 2012*; *Li et al., 2014*; *Duncan et al., 2015*). Using a multiple linear regression approach, we found that *FLM-ß* and *FLC* explained 32.9% ($R^2 = 0.329$, p=0.0083) of flowering time variation, with *FLM-ß* significantly explaining a subfraction of 21.0% ($R^2 = 0.210$, p=0.011) and *FLC* only 11.9%, however not significantly ($R^2 = 0.119$, p>0.05) (*Figure 8B* and *Figure 8—figure supplement 2B*). Further, by integration of expression data and flowering time data into a multiple linear model and comparison of the slopes, we found that flowering time responded stronger to *FLM-ß* levels in the accessions than in the transgenic variants (p=0.0321) (*Figure 5B* and *Figure 8*). Taken together, we concluded that variations in *FLM-ß* levels account for flowering time in cool ambient spring temperatures in a diverse population of summer-annual Arabidopsis accessions (*Figure 9*).

## Discussion

Ambient temperature during spring is a major cue determining flowering time. Cool temperatures generally delay and warm temperatures promote flowering time of Arabidopsis. The *FLM* locus explains flowering time variation in different ambient temperatures but the underlying genetic bases of *FLM*-dependent flowering remained largely unclear (*Salomé et al., 2011*; *el-Lithy et al., 2006*; *O'Neill et al., 2008*).

We found that, besides the *FLM* promoter, also intron 1 sequences were essential for *FLM* basal expression and subsequently identified by phylogenomic footprinting a conserved 373 bp intron 1 region essential for *FLM* basal expression (*Figure 2*). Further, through association analyses using genomic sequence information, we uncovered *FLM* regulatory regions (PRO2+ and INT6+) that control temperature-dependent *FLM* expression in a haplotype-specific manner (*Figure 4*). While most

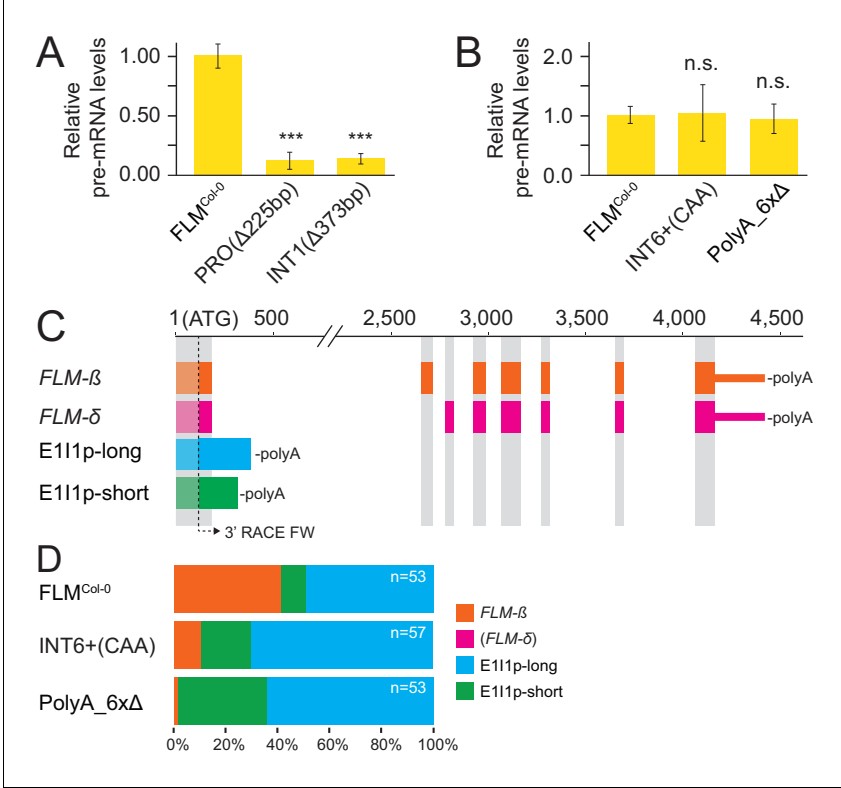

**Figure 7.** *FLM* polymorphisms affect *FLM* transcription or splicing at the expense of *FLM-ß*. (**A**) and (**B**) Mean and SD (n = 3) of *FLM* pre-mRNA levels. Student's t-test: ***p≤0.001; n.s. = not significant. (**C**) Schematic representation of *FLM* cDNAs detected more than once (n = 163). *FLM-δ* transcripts were not detected and are only shown for completeness. Grey areas correspond to *FLM* exons of the *FLM-ß* and *FLM-δ* gene models as specified in *Figure 1A*. The arrow indicates the position of the 3' RACE primer used in combination with an oligo (dT) reverse primer to detect polyadenylated transcripts. (**D**) Frequency distribution of the transcripts displayed in (**B**) with total number of sequences depicted in the graph.

The following figure supplement is available for figure 7:

**Figure supplement 1.** Transgenic lines expressing different *FLM* alleles display differential *FLM* expression and splicing patterns.

---

PRO2+ and INT6+ haplotypes displayed differential basal but temperature-sensitive *FLM* expression, correlating strongly with changes in flowering time, one INT6+ variant, INT6+$^{CAA}$, was strongly compromised in temperature-sensitive *FLM* regulation and flowering pointing at the importance of this intronic region for flowering time adaptation (*Figures 4* and *6*).

In all our experiments, *FLM-ß* highly correlated ($R^2$ = 0.94) with flowering time over a broad vegetative range (15–30 rosette leaves) when determined at 15°C and tested in a homozygous background, regardless of the type of variant (*Figure 5*). Our finding that *FLM-ß* had a stronger effect in the control of flowering time than *FLM-δ* indicates that previously phrased functional models proposing that FLM-δ had an antagonistic activity to FLM-ß need to be corrected (*Posé et al., 2013*; *Sureshkumar et al., 2016*). This is further supported by the fact that our results estimate that *FLM-δ* levels are overall very low and that we did not identify a single *FLM-δ* clone in a directed sequencing approach that identified 53 *FLM-ß* clones. Our findings thus support more recent studies suggesting that the *FLM-δ* splice variant may be biologically irrelevant (*Sureshkumar et al., 2016*; *Lutz et al., 2015*). One of these studies also identified a large number of biologically irrelevant transcripts that could be identified with the primer combination used here for the detection of *FLM-δ* (*Sureshkumar et al., 2016*). Our data thus support the conclusion that none of these amplification products has a biologically important function (*Sureshkumar et al., 2016*).

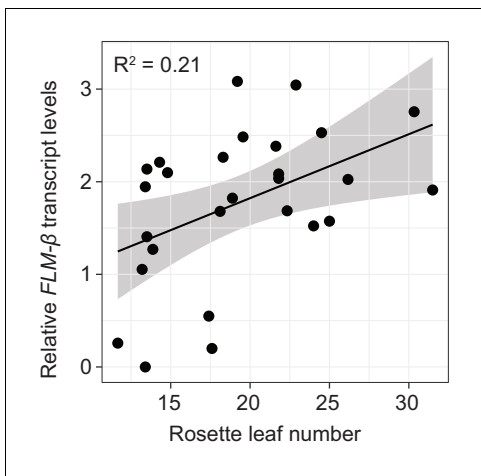

**Figure 8.** *FLM-ß* expression shows high correlation with flowering time in Arabidopsis accessions. (**A**) Correlation (simple linear regression) of *FLM-ß* transcript levels detected in transgenic plants and natural accessions with identical PRO2+ and INT6+ haplotypes. When the contribution of individual haplotypes was assessed, PRO2+[AAAAC] INT6+[CAA] showed an exceptional response in accessions and was excluded. *Figure 8—figure supplement 1E* shows the complete analysis. Horizontal and vertical error bars depict SD of four replicate transgenic line pools or SD of the accessions. Data of the transgenic lines was taken from *Figure 4E*. The grey area indicates the 95% confidence interval. (**B**) Correlation of *FLM-ß* transcript levels with flowering time (n = 8–10) as measured in rosette leaf number of summer-annual accessions. p=0.011.

The following figure supplements are available for figure 8:

**Figure supplement 1.** Geographic and genetic distribution of the PRO2+/INT6+ haplotypes among 840 accessions.

**Figure supplement 2.** Geographic distribution of summer-annual accessions.

Whereas the PRO2+ and INT6+ polymorphisms tested affected *FLM* transcript abundance, the molecular causes of these transcription changes varied among the different polymorphisms (*Figures 4* and *7*). Deletion of a 225 bp promoter region including the PRO2+ polymorphic site was associated with tenfold reduced levels of *FLM* pre-mRNA indicating that this promoter region was essential for *FLM* expression (*Figure 7*). A twofold relative difference of *FLM-ß* levels was found when comparing the PRO2+[GATAC] and PRO2+[AAACC] variants with the lowest and highest *FLM-ß* expression, respectively, and our linear model would predict a flowering time difference of 14.4 leaves at 15°C (*Figures 4* and *5*). The PRO2+ region harbours a predicted MADS-box transcription factor binding site and several instances of auto- or cross-regulation of MADS-box factors have been described (*de Folter et al., 2005*; *Kaufmann et al., 2009*; *Smaczniak et al., 2012*). It can therefore be envisioned that *FLM* expression is regulated by MADS-box transcription factors and that PRO2+ polymorphisms modulate the efficiency or specificity of these binding events and thereby modulate basal *FLM* expression.

Similarly, the deletion of a 373 bp fragment in *FLM* intron 1 resulted in an elevenfold reduction in *FLM* expression and earlier flowering by 11.5 leaves as experimentally determined. This effect size is much smaller than predicted by the linear model. However, as proposed earlier, this may likely be due to a critical lower threshold for *FLM-ß* to become effective (*Figures 4* and *5*). Intronic *cis*-regulatory transcription factor binding sites have been identified in other MADS-box transcription factors and interactions of enhancer and silencer elements that reside in the promoter sequence or the 3'-end of the first intron were reported (*Hong et al., 2003*; *Schauer et al., 2009*). Thus, similar mechanisms may govern the expression of *FLM* at intron 1 sites that may act in isolation or together with binding events at the *FLM* promoter.

Interestingly, natural polymorphisms in intron 6 (INT6+) led to about fourfold differences in the abundance of *FLM-ß*, which could, as predicted by the linear model, relate to a flowering time delay by 28.8 leaves, suggesting that INT6+ harbours extensive potential to fine-tune flowering (*Figures 4* and *5*). Importantly, this molecular effect appears to be mediated, in the case of INT6+[CAA], by effects on the splicing efficiency and specificity at the distal intron 1. There, INT6+[CAA] promotes the formation of short intron 1-containing transcripts, at the expense of *FLM-ß*, that are likely subjected for degradation by nonsense-mediated decay (*Figure 7*) (*Lutz et al., 2015*).

The INT6+ site is directly flanked by a short PolyA motif and such sites represent potential recognition sites of hnRNP (heterogeneous nuclear ribonucleoprotein) splicing factors. The predicted hnRNP binding pattern at INT6+ depended indeed on the INT6+ haplotype, when predicted by web-based algorithms, and their binding preference and activity, in concert with other splicing or

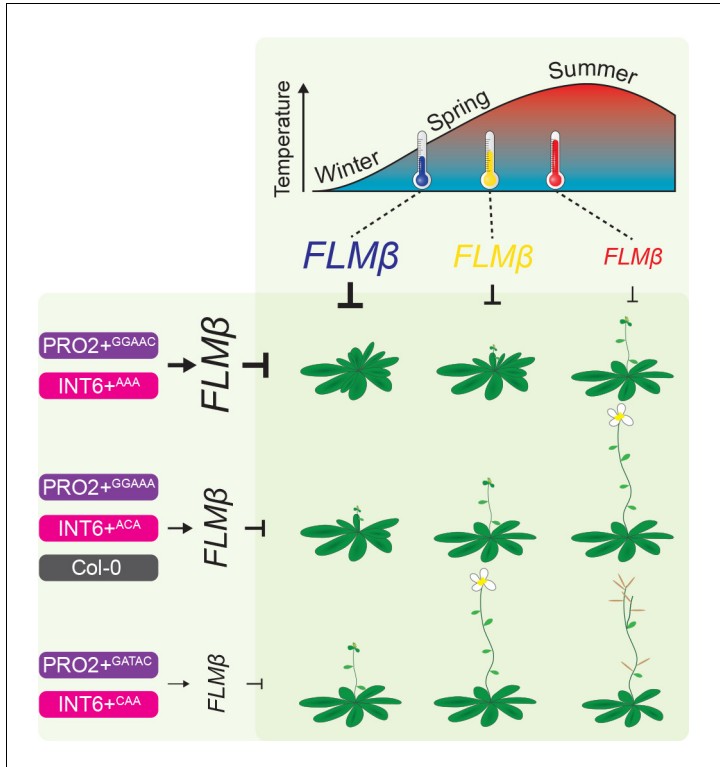

**Figure 9.** Model of the proposed role of PRO2+ and INT6+ haplotypes and temperature on *FLM-ß* abundance and flowering. The abundance of the flowering repressor *FLM-ß* decreases in response to higher temperature and flowering is consequently accelerated (*Figures 1B* and *4E*) (*Lee et al., 2013*; *Posé et al., 2013*; *Lutz et al., 2015*). Note, that previous studies showed an especially prominent effect of *FLM* in a range from 9°C to 21°C (long-day photoperiod) (*Lee et al., 2013*; *Posé et al., 2013*; *Lutz et al., 2015*). At the genetic level, *FLM-ß* abundance is triggered by the PRO2+ (purple) and/or the INT6+ (pink) haplotype and flowering time correlated to *FLM-ß* abundance (*Figure 4E*). Among the PRO2+/INT6+ combinations tested, the Col-0 (grey) reference allele (PRO2 +$^{AAAAC}$/INT6+$^{AAT}$) showed intermediate *FLM-ß* levels (*Figure 4E*). We suggest, that changes in flowering time due to changing ambient temperature can be precisely buffered by modifying the PRO2+ and INT6+ regions, as illustrated by similar plant symbols.

transcriptional regulators, may ultimately be the basis of the splicing changes observed here (*Piva et al., 2012*; *Carrillo Oesterreich et al., 2011*; *Reddy et al., 2013*).

The cooccurrence of PolyA motifs ([A]$_{7-11}$) with the PRO2+ and the INT6+ sites had attracted our attention (*Figure 4—figure supplement 1*). Combinatorial deletions of all six *FLM* PolyA motifs led to gradual decreases in *FLM-ß* abundance, which could be explained be altered intron 1 splicing (*Figure 7*). This ultimately resulted in *FLM-ß* levels below a lower effective threshold in the Poly-A_6xΔ lines and consequently very early flowering (*Figure 4—figure supplement 1*). This suggested that the PolyA motifs may modulate *FLM* expression by altering *FLM* splicing. In support of this conclusion, we found that an extended PolyA motif, as it is present in the INT6+$^{AAA}$ variant when compared to the Col-0 reference variant INT6+$^{AAT}$, correlated with strong increases in *FLM-ß* but not in *FLM-δ* abundance (*Figure 4*). PolyA motifs are known to prevent nucleosome binding and changes of chromatin architecture may influence splicing (*Reddy et al., 2013*; *Suter et al., 2000*; *Wijnker et al., 2013*). The knowledge about the underlying molecular mechanisms and the identity and specificity of the splicing regulators in plants is still very limited. We noted with interest, however, that several hnRNPs have a reported role in flowering time regulation in Arabidopsis and wheat (*Kippes et al., 2015*; *Fusaro et al., 2007*; *Streitner et al., 2012*; *Xiao et al., 2015*).

The nine PRO2+/INT6+ haplotype combinations included in our transgenic experiments represent 69% of the world-wide PRO2+/INT6+ variation (*Figure 8—figure supplement 1A,B*). We found, that *FLM-ß* explained around 21% of flowering time in a genetically heterogeneous

population of summer-annual Arabidopsis accessions (*Figure 8*). Hence, PRO2+ and INT6+ haplotypes regulate *FLM-ß* abundance and, in turn, contribute to flowering time regulation in Arabidopsis accessions (*Figures 8* and *9*). We consider it not surprising that the correlation of flowering time with *FLM-ß* levels (21% versus 94%) as well as the responsiveness of flowering time to *FLM-ß* levels differ between a genetically heterogeneous natural population and the homogenous transgenic population. First, we showed that residual *FLC* transcript may also slightly contribute to variation of flowering time of summer-annual accessions, possibly interfering with the effects of *FLM-ß* (*Balasubramanian et al., 2006*; *Li et al., 2006*). Further, additional sequence variation in *FLM* in natural accessions may contribute to relevant expression changes, e.g. in the part of intron 1 identified as essential for *FLM* expression but not analysed in further detail here (*Figure 2*). Then, it is possible that these accessions harbour variation in the *FLM*-related *MAF2 - 4*, and this variation may differentially affect flowering time, in isolation or in combination with variation of their common interaction partner *SVP* (*Ratcliffe et al., 2003*; *Airoldi et al., 2015*; *Scortecci et al., 2003*). Bearing these possible genetic and environmental interferences in mind, we regard the detected effect of *FLM-ß* on flowering (21%) in cool (15°C) temperatures as considerable and suggest that it is rather an underestimation.

We found a uniform distribution of nine PRO2+/INT6+ haplotype combinations among the genetic clusters that have recently been established based on the analysis of genomic sequences from 1135 Arabidopsis accessions (*The 1001 Genomes Consortium, 2016*). Interestingly, the PRO2 $+^{GGAAC}$/INT6$+^{AAT}$ haplotype was overrepresented among the relict accessions and the PRO2+-$^{AAAAC}$/INT6$+^{AAA}$ haplotype was overrepresented among the Asian group (*The 1001 Genomes Consortium, 2016*) (*Figure 8—figure supplement 1*). In our experiments, both of these haplotypes were associated with increased *FLM-ß* transcript levels and consequently late flowering (*Figure 4*). Since the relict and Asian accessions have been proposed to originate from glacial refugia from where central Europe was recolonized after the last ice age, these late-flowering PRO2+/INT6+ haplotypes may represent original haplotypes (*Sharbel et al., 2000*; *Schmid et al., 2006*). The late flowering phenotype associated with these haplotypes would be in line with the hypothesis that late flowering alleles are more ancient and that early flowering alleles were derived only during the more recent Arabidopsis evolution (*Toomajian et al., 2006*). Further, genetic linkage among the PRO2+ and INT6+ polymorphic sites was overall low ($R^2 = 0.11–0.55$) and the sequences surrounding PRO2+ and INT6+ were comparatively conserved. Taken together, this suggests that mutations in the PRO2+ and INT6+ sites independently arose multiple times and that these sites were preferentially selected during Arabidopsis flowering adaptation. All PRO2+/INT6+ alleles displayed a broad geographic distribution, except PRO2$+^{GGAAC}$/INT6$+^{AAT}$, which was overrepresented in Spain (*Figure 8—figure supplement 1*). Likely, microclimates at specific geographic locations rather than general climate conditions at broad geographic regions may be important to drive adaptation of flowering to changing ambient temperature, as it was previously suggested for broadly distributed non-coding haplotypes of *FLC* that explain variation in vernalization (*Li et al., 2014*; *Weigel, 2012*; *Shindo et al., 2006*).

Mutations in cis-regulatory regions are important for adaptation and phenotypic evolution and have a low probability to generate negative pleiotropic effects (*Swinnen et al., 2016*; *Meyer and Purugganan, 2013*; *Wray, 2007*). *FLM* may be an ideal candidate for flowering time adaptation through non-coding sequence variation at PRO2+ and INT6+ sites since changes in *FLM-ß* abundance precisely modulate flowering while maintaining phenotypic plasticity and without generating negative pleiotropic effects (*Figure 9*). Changes at the PRO2+ and INT6+ sites should allow adapting flowering time in response to altered geographic distribution and consequently climate conditions as well as during changing global environments (*Figure 9*). The role of *FLM* orthologues in other *Brassicaceae*, including a number of agronomically important species, is as yet not understood. It is thus at present not possible to predict the role of Arabidopsis *FLM* polymorphisms for flowering time adaptation and plant breeding in this plant family. However, in view of the availability of new genome editing methodologies, the knowledge about important non-coding regions may be useful to fine-tune flowering time (*Shan et al., 2013*). This may allow buffering the anticipated negative effects on agricultural systems that occur as a consequence of small temperature changes during climate change (*Moore and Lobell, 2015*; *Wheeler and von Braun, 2013*).

## Materials and methods

### Biological material

All *Arabidopsis thaliana* accessions used in this study as well as *flm-3* (Salk_141971; Col-0) and Nd-1 (N1636) were provided by the Nottingham Arabidopsis Stock Centre (NASC; Nottingham, UK). *flc-3* and $FRI^{SF-2}$ *FLC* (*Michaels and Amasino, 1999*) were a gift from Franziska Turck and George Coupland (Max-Planck Institute of Plant Breeding Research, Cologne, Germany). pFLM::gFLM (pDP34), pFLM::FLM-ß (pDP96), and pFLM::FLM-δ (pDP97) were previously reported (*Posé et al., 2013*). Primers to screen the accessions for the $FLM^{LINE}$ insertion were used as previously described (*Lutz et al., 2015*).

### Physiological experiments

For flowering time analyses, plants were grown under long day-conditions with 16 hr white light (110–130 µmol m$^{-2}$ s$^{-1}$)/8 hr dark in MLR-351 SANYO growth chambers (Ewald, Bad Nenndorf, Germany). Plants were randomly arranged in trays and trays were rearranged every day. Water was supplied by subirrigation. Flowering time was quantified by counting rosette leaf numbers (RLN). Consistent with previous reports, we observed a strong correlation between days to bolting and rosette leaf number of Arabidopsis accessions (*Figure 8—figure supplement 2C*) (*Atwell et al., 2010*). Student's t-tests (normally distributed values), Wilcoxon rank tests (not normally distributed values), and multiple regression models were calculated with R (http://www.r-project.org/).

### Cloning procedure

A previously described construct with the Col-0 genomic *FLM* fragment pFLM::gFLM template (pDP34) (*Posé et al., 2013*) was recombined into a pDONR201 destination vector using the Gateway system (Life Technologies, Carlsbad, CA). This vector was used as a template (FLM$^{Col-0}$) to generate mutations using either a single phosphorylated primer or a combination of forward and reverse primers (*Sawano and Miyawaki, 2000*; *Hansson et al., 2008*). In case of variants with multiple modifications, individual mutations were introduced one at a time. The mutated inserts were recombined to the pFAST-R07 expression vector using the Gateway system (Life Technologies, Carlsbad, CA) (*Shimada et al., 2010*). All expression constructs were verified by sequencing and transformed into *Agrobacterium tumefaciens* strain GV3101. Nd-1 plants were transformed using the floral-dip method (*Clough and Bent, 1998*) and transgenic plants were identified based on seed fluorescence (*Shimada et al., 2010*). Segregation of T2 lines was examined and lines with single insertion events were selected for further analysis based on segregation ratios (*Shimada et al., 2010*). A list of primers and expression constructs is listed in *Supplementary file 7*.

### Quantitative real-time PCR

For qRT-PCR analyses of homozygous lines, total RNA was isolated from three biological replicates using the NucleoSpin RNA kit (Machery-Nagel, Düren, Germany). DNA was removed by an on-column treatment with rDNase (Machery-Nagel, Düren, Germany). 2–3 µg total RNA were reverse transcribed with M-MuLV Reverse Transcriptase (Thermo Fisher Scientific, Waltham, USA) using an oligo (dT) primer. The cDNA equivalent of 30–50 ng total RNA was used in a 12 µl PCR reaction with SsoAdvancedUniversal SYBR Green Supermix (BioRad, München, Germany) in a CFX96 Real-Time System Cycler (BioRad, München, Germany). The relative quantification was calculated with the *ΔΔ*Ct method using *ACT8* (AT1G49240) as a standard (*Pfaffl, 2001*). For the analysis of the pooled independent T2 transgenic lines, a quarter of all available independent lines per construct (18 to 45) was sampled resulting in four replicate pools comprising between four and eleven lines. Around 1000 seeds were used for pooling per line. One RNA sample was extracted per replicate pool and processed as described above.

The large scale expression experiments were performed using a previously described 96-well format (*Figure 8* and *Supplementary file 3*) (*Box et al., 2011*). DNA was digested with DNaseI (Thermo Fisher Scientific, Waltham, USA) and reverse transcription and qPCR reactions were performed as described above using a CFX384 Real-Time System Cycler (BioRad, München, Germany). *ACT8* (AT1G49240) and *BETA-TUBULIN-4* (AT5G44340) were used as reference genes. Student's t-tests were calculated with Excel (Microsoft). qRT-PCR primers are listed in *Supplementary file 7*.

## Data retrieval from the 1001 *Arabidopsis thaliana* genome project

Sequence information from datasets of 776 and 840 accessions, respectively, have been available that were used in this study. The genomic sequences of a comprehensive set of 1135 Arabidopsis accessions have just recently been released and were used for concluding analyses (*The 1001 Genomes Consortium, 2016*). Genomic *FLM* sequences (7 kb; Chr1: 28953637–28960296; including 2 kb upstream and 0.2 kb downstream sequence) were extracted from 776 accessions from the 1001 Genomes portal (http://1001genomes.org/datacenter/) using the Wang dataset (343 accessions), the GMI dataset (180 accessions), the Salk dataset (171 accessions) and the MPI dataset (80 accessions) (*The 1001 Genomes Consortium, 2016*; *Schmitz et al., 2013*; *Long et al., 2013*; *Cao et al., 2011*). 45 SNPs with a MAF>5% were extracted and used for haplotype analysis of the *FLM* locus (*Figure 3A* and *Figure 3—figure supplement 1*). 31 SNPs were polymorphic between the 54 selected accessions represented in the *FLM* haplotype set (*Supplementary file 1*).

To obtain an extended set of sequences that included not only SNPs but also information about insertions and deletions, we extracted *FLM* genomic sequences (including 2 kb upstream and 0.38 kb downstream sequence) from the GEBrowser 3.0 resource (http://signal.salk.edu/atg1001/3.0/gebrowser.php). A set of 850 sequences was manually curated and aligned using MEGA7.0.14 (*Tamura et al., 2011*). Some sequences were excluded since they possessed a high number of ambiguous bases. The core sequence set consisted of 840 sequences. This sequence set was used for association analysis (*Figure 3—figure supplements 4–7*) and all further analysis on *FLM* PRO2+ and INT6+ variation (*Figure 8* and *Figure 8—figure supplement 1*). Accession identifiers, geographic data, and ADMIXTURE group membership (k = 9) were obtained from the 1001 genomes tool collection (http://tools.1001genomes.org/) (*The 1001 Genomes Consortium, 2016*).

## Haplotype analysis

Haplotype analyses were performed with DNaSP 5.10 (*Rozas and Rozas, 1995*) using the SNP dataset of the above-described set of 776 sequences. Invariable sites were removed and networks were generated using FLUXUS network software (*Bandelt et al., 1999*).

## Phylogenetic tree and alignments

Alignments were calculated using ClustalW2 or MUSCLE and Neighbor-Joining trees (Maximum Composite Likelihood method, 1000 bootstrap replicates) were constructed with Geneious vR7.0.5 (Biomatters Limited, Auckland, New Zealand) and MEGA7.0.14 (*Tamura et al., 2011*).

## LD analysis

Linkage ($R^2$) between the polymorphic sites that were used as input for the simple single locus association test was calculated using the LD function in GGT v2.0 (*van Berloo, 2008*). Only diallelic SNPs (109 of 119) were considered.

## Kruskal-Wallis test

To detect variants with significant effects on *FLM* transcript levels (simple single locus association test), 119 variants of a 7 kb *FLM* locus were extracted from a set of 840 sequences retrieved from the GEBrowser 3.0, as described above. To examine whether a variant showed a significant effect on *FLM* expression, the qRT-PCR expression values of the 54 accessions from the *FLM* haplotype set were used as input data (*Supplementary file 4*) to run Kruskal-Wallis tests using R (http://www.r-project.org/). p-values from all comparisons were corrected following the Benjamini-Hochberg multiple testing correction and resulting values were -log(10) transformed and plotted along the gene model (*Figure 3—figure supplement 5* and *Supplementary file 5*).

## Phylogenetic footprinting

*FLM* orthologues were identified first by OrthoMCL (V2.0) clustering (PMID: 12952885; standard parameters, inflation value = 1.5, BLAST e-value-cutoff = e-05), incorporating predicted protein sequences from *Arabidopsis thaliana* (TAIR10; AT1G77080.4, AT5G65060.1), *Arabidopsis lyrata* (V1.0; fgenesh2_kg.2__2000__AT1G77080, fgenesh2_kg.8__2543__AT5G65050, fgenesh2_kg.233__4__AT5G65050), *Boechera stricta* (V1.2; Bostr.4104s0001.1), *Brassica rapa* (V1.3; Brara.B03928.1.p, Brara.F02378.1.p), *Brassica oleracea* (V2.1; Bo2g166500.1, Bo2g166560.1), *Capsella*

*grandiflora* (V1.1; Cagra.0450s0030.1, Cagra.0917s0081.1), *Capsella rubella* (V1.0; Carubv10020979m, Carubv10027327m), *Gossypium raimondii* (V2.1), *Medicago truncatula* (V4.0), *Oryza sativa* (MSU7), *Populus trichocarpa* (V3.0) and *Solanum lycopersicum* (ITAG2.3). Data sets were downloaded from Phytozome (*Goodstein et al., 2012*), PGSB PlantsDB (*Spannagl et al., 2016*) and Ensembl Plants (*Kersey et al., 2016*). Cluster(s) containing *FLM* were extracted and, in case of the presence of multiple group members from individual species, filtered further using the best bi-directional BLAST hit criterion. Genomic sequences and 2 kb upstream region were aligned with ClustalW2 (*Larkin et al., 2007*). Conserved regions in the alignment were manually annotated.

### *FLM* molecular analyses

qRT-PCRs for *FLM* pre-mRNA abundance and 3' RACE PCR were performed as previously described (*Lutz et al., 2015*). Primers sequences are listed in *Supplementary file 7*.

## Acknowledgements

The authors would like to thank Detlef Weigel (MPI Developmental Biology, Tübingen, Germany), Markus Schmid (MPI Developmental Biology, Tübingen, Germany; Umea Plant Science Center, Umea, Sweden) and David Posé (MPI Developmental Biology, Tübingen, Germany; University of Malaga, Spain) for materials and discussions. We thank Thomas Regnault and Alexandre Magalhães for discussion and critical reading of the manuscript. Paula Thompson (Technical University of Munich) is thanked for language editing. This work was supported by grants from the Deutsche Forschungsgemeinschaft as part of the SPP1530 'Flowering time control: from natural variation to crop improvement.' to Claus Schwechheimer and as part of the Sonderforschungsbereich 924 'Molecular mechanisms regulating yield and yield stability' to Claus Schwechheimer and Klaus FX Mayer.

## Additional information

### Funding

| Funder | Grant reference number | Author |
|---|---|---|
| Deutsche Forschungsgemeinschaft | SPP1530 | Claus Schwechheimer |
| Deutsche Forschungsgemeinschaft | SFB924 | Klaus FX Mayer Claus Schwechheimer |

The funders had no role in study design, data collection and interpretation, or the decision to submit the work for publication.

### Author contributions

UL, Conceptualization, Data curation, Formal analysis, Investigation, Visualization, Methodology, Writing—original draft, Writing—review and editing; TN, MS, JD, Data curation, Formal analysis; KFXM, Data curation, Supervision, Funding acquisition, Methodology; CS, Conceptualization, Supervision, Funding acquisition, Project administration, Writing—review and editing

### Author ORCIDs

Ulrich Lutz, http://orcid.org/0000-0003-3625-3440
Claus Schwechheimer, http://orcid.org/0000-0003-0269-2330

## Additional files

### Supplementary files

• Supplementary file 1. List of 45 SNPs for haplotype analysis.

• Supplementary file 2. Identifier and geographic information of the 54 *FLM* haplotype set accessions.

• Supplementary file 3. Expression data of 54 *FLM* haplotype set accessions.

• Supplementary file 4. Input data for association analysis.

• Supplementary file 5. Output data from association analysis (p-values).

• Supplementary file 6. Expression data of 85 (A) and 27 (B) accessions.

• Supplementary file 7. List of primers used in this study for cloning (A) and qRT, pre-mRNA quantification, and 3' RACE PCRs (B).

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
