## [Decision Letter]

Thank you for submitting your article "Natural haplotypes of FLM non-coding sequences fine-tune flowering time in ambient spring temperatures in *Arabidopsis*" for consideration by *eLife*. Your article has been favorably evaluated by Ian Baldwin (Senior Editor) and three reviewers, one of whom is a member of our Board of Reviewing Editors. The following individual involved in review of your submission has agreed to reveal his identity: Dan Runcie (Reviewer #2).

The reviewers have discussed the reviews with one another and the Reviewing Editor has drafted this decision to help you prepare a revised submission.

The manuscript describes an interesting illustration of how to utilize genome wide association in combination with a variety of genomic tools and thoughts to begin to improve the model of how FLM integrates temperature into the flowering time decision.

There are four key concerns that need to be addressed prior to any final decision.

1) The FLM expression to flowering correlation is not sufficient. The exclusion of multiple accessions was considered not justified. A multiple regression approach with FLC expression included in the model is a way that may account for this and would be a more correct statistical test of the hypothesis.

2) There are other significant statistical concerns that may affect the core claims from the manuscript such as temperature sensitivity, etc. These must be corrected and the claims either supported or changed pending the results.

3) The role of vernalization in FLM expression and variation thereof needs to be assessed to allow for the data to be properly integrated into the broader model. Given that the integration into the broader model is the critical interest in this paper.

4) There was concern about including FLMd if it is not found in the measurements? This should be addressed.

Essential revisions:

Please also address the following concerns that will improve the manuscript.

*Reviewer #2:*

The authors present a detailed search for the genetic basis of variation in ambient temperature sensitivity for flowering time in *Arabidopsis*. Focusing specifically on the control of the expression of FLM which has been previously linked to temperature sensitivity, the authors draw on a range of modern and innovative tools to pinpoint alleles conferring variation FLM expression within the species. They identify the major alleles controlling both basal expression and temperature sensitivity of FLM segregating at high frequency in the species and confirm effects of these alleles using transgenics. Overall, this study is important in its demonstration of how modern tools in *Arabidopsis* can be used to rapidly identify the specific base-pairs underlying natural variation in important traits. While I think that the major conclusions seem robust and well-supported, this is a very complicated paper and I do have concerns about specific analyses and interpretations.

1) My main concern is with the central focus on the comparison between two isoforms of FLM (FLMb and FLMd). A recent study showed that the qPCR assays targeting these isoforms actually pick up other isoforms, many of which produce truncated proteins (Sureshkumar et al. 2016 Nature Plants). Here, the authors note that they can't even find a true FLMd transcript (out of 163 tested). It is not clear what if any conclusions can be drawn from the FLMd assay, and so devoting so much space throughout the article and figures to FLMd is not useful and distracting. On the other hand, this and earlier studies the FLMb assay does seem to be useful for assaying FLM activity, so I am comfortable with these results.

2) Secondly, I am concerned with the results presented in Figure 8 – the correlations between FLM expression in accessions and a) expression in constructs and b) flowering times of the accessions. In order to find a correlation between FLM expression in accessions and in the transgenic lines with the corresponding constructs, the authors dropped an "outlier" point (1 each for FLMb and FLMd). No justification for this is given beyond the point not fitting expectations. It's not clear from Figure 8—figure supplement 1 that these points are really more outliers than any others. And, it's not that this is really a single point – on the y-axis this is the average expression of ~10 accessions, so why should they all be dropped?

3) For the flowering time comparison, the lines were vernalized. However, FLM is known to be repressed by vernalization (as the authors note), so how could it be affecting flowering, and might there be variation among lines in how FLM responds to the vernalization? It seems more likely that other loci that are correlated with FLM are controlling flowering time here. The authors should either measure FLM expression in these vernalized plants and correlate that with flowering, or repeat the experiment without vernalization where FLM and FLC expression in these lines has already been measured. A better way to control for FLC variation would be to statistically account for its effects with multiple regression.

4) Finally, on the statistical side, the conclusions about "temperature insensitivity" for INT6+ alleles are not backed by statistics – the authors should explicitly test for a change in the temperature sensitivity between FLM-Col0 and INT6+. From the graphs, it appears temperature sensitivity is reduced, but certainly not eliminated (Figure 6).

*Reviewer #3:*

In “Natural haplotypes of FLM non-coding sequences fine-tune flowering time in ambient spring temperatures in *Arabidopsis*" Lutz et al. address the biological relevance of FLM splice variants for variation in flowering time in natural accessions and identify non-coding sequence polymorphisms that contribute to the regulation of basal expression levels and the differential, thermo-responsive generation of the β and δ splice variants. As such, this work provides novel insights into the regulation of flowering in the ambient temperature range and manages to dissect polymorphisms which are relevant for basal FLM expression and temperature-induced effects on the β and δ splice variants.

The manuscript is overall very well written and the authors provide extensive data which has apparently been subjected to comprehensive and sound statistical analyses. The data is presented in an attractive manner. Main manuscript figures are supported by extensive supplementary material including the presentation of extensive control data, which is greatly appreciated. In some cases the figure descriptions or legends should be extended. Due to the vast amount of data condensed in the figures, the descriptions (at least in the text) should be detailed enough for a wide readership to follow the authors through their argumentation.

1) The authors state that “The presence of intron 1 is sufficient to restore the basal expression of FLM” – while it does restore expression levels to detectable levels, the expression level of the β splice variant at 15°C is considerably lower than the WT, whereas the FLMδ levels are similar the WT at this temperature. This would indicate that intron 1 alone cannot fully account for basal levels of FLM β.

2) The use of pools of independent segregating T2 lines for transcriptional analysis seems unusual. While the controls presented in Figure 2—figure supplement 2 suggest that this can recapitulate the “natural” situation (why not subjected to statistical analysis?), it seems nonetheless risky to deliberately tolerate the presence of non-transgenics in the mix. Could the authors state their reasoning for this unusual procedure and provide the missing statistics here? Also, the phrase “replicate pools compromising…” should probably read “comprising”.

3) I had a hard time to comprehensively grasp the information provided in Figure 3, especially when it comes to the color code and its use in Figure 3. Figure 3—figure supplement 1 provides additional information, but still the color coding of haplotypes which is provided for three different regions (full length, intron 1 or intron 6) is confusing as the composition of the different haplotypes varies depending on the selected gene region (Figure 3—figure supplement.1G). The authors may want to better explain these figures to enable a wide readership to follow their analysis and key points here. Also, in the caption of Figure 3—figure supplement 1 the haplotype “Numbers on the left” probably refer to (D) not (E)?

4) Figure 5: Please specify again in the figure caption that this data corresponds to 40 selected accessions. Why was the FLM β level not determined in the vernalized plants? Even though FLC effects seem to be generally negligible, it would have been as easy to do the analysis on the same material.

5) In Figure 5—figure supplement 1: removing the lower quartile (or more!) of flowering time data for the correlation analysis by simple assumption that these represent the non-transformed Nd-1 individuals of the segregating population is questionable. While this may be an attractive assumption, it negates the fact that each line pool may represent its individual set of variances (also present in ND-1 itself). This would at least require a spot check for the presence of transgenes in this pool or the percentage of “clean” Nd-1 individuals. In principle, a simple test by PCR would have provided a solid answer to that. As is, the authors should include these lines in the correlation analysis as the factual genotype is not known.

[Editors' note: further revisions were requested prior to acceptance, as described below.]

Thank you for resubmitting your work entitled "Natural haplotypes of FLM non-coding sequences fine-tune flowering time in ambient spring temperatures in *Arabidopsis*" for further consideration at *eLife*. Your revised article has been favorably evaluated by Ian Baldwin (Senior Editor), a Reviewing Editor, and two reviewers.

The manuscript has been improved but there are some remaining issues that need to be addressed before acceptance, as outlined below:

There are still some key issues on interpretation and analysis that need to be addressed. None of these require new wet lab experiments and are relatively easy and quick to clean up. All felt that this is necessary.

*Reviewer #2:*

I thank the authors for their improvements to the paper – the figures especially are more clear now. I still have the following concerns about the overall conclusions and analysis. These relate to three important conclusions:

1) That FLM variation contributes to flowering time variation across a diverse panel of summer annual accessions.

The new experiment to test this is very helpful, and does provide good evidence that FLMB expression level is important for flowering time variation. My concern is with interpretation: The authors provide several explanations for why the correlation between flowering and FLMB expression in this accession set is lower than in the transgenics. This is trivial because there is much more variation within each haplotype class in the accessions than the transgenics, because they vary at many loci across the genome. I think the real question should be: Is the slope of the relationship between FLMB and flowering time significantly less in the accessions? This would ask if the equivalent change in FLMB expression would cause the same change in flowering. This is a more meaningful metric than change in R2. The correct test would be to include both accessions and transgenics in the same model (flowering_time ~ FLMB * genotype_class), and ask if the interaction is significant.

2) That the 9 promoter haplotypes explain the variation in FLMB expression in the accessions.

This is the aim of the analysis in Figure 8 and Figure 8—figure supplement 1. The main claim is that the expression variation induced by these haplotypes in the transgenics is correlated with the expression variation of accessions carrying the same haplotypes. This conclusion only holds when one of the 9 haplotypes is excluded. The explanation given for excluding this haplotype is that the correlation goes up after removing this. I did a quick simulation study and found that using this algorithm you'd expect that the best correlation found by dropping each of the 9 pairs would be greater than 46% about 21% of the time even if there were actually no correlation at all. So, this is really not very strong evidence that the haplotype effects are similar. Certainly, the statement "We found that the PRO2+/INT6+ effects, as detected in transgenic experiments (R2 = 0.94), partially explain the FLM-ß expression variation in natural *Arabidopsis* accessions (R2 = 0.46)" is not supported, because this 46% number is only true when you exclude those accessions that don't fit.

As above, I think a more appropriate analysis would be to ask if the sizes of the differences between the haplotypes is the same in the transgenics and the accessions (i.e., is the slope of the graph different from 1?). A more straightforward answer to the question of how much variation in FLMB is explained by these haplotypes is simply to report the R2 for the analysis shown in Figure 8—figure supplement 1.

3) That the INT6+(CAA) haplotype controls temperature sensitivity.

This conclusion is important because of the model that FLM controls temperature sensitivity of flowering. New data are presented here relative to the previous version, though where they came from is not clear. The correct test for a change in temperature sensitivity is to ask if the interaction between genotype and temperature is significant (expression ~ Genotype + Temp + Genotype:Temp). The n.s. effect of temperature on expression for INT6+ is not evidence of no temperature effect, only a lack of evidence for a temperature effect. The figure caption states the statistics are done based on a t-test. A t-test can't be used to test for an interaction (except in 6C/D if the 5 transgenic lines per genotype were used as the replicates, n = 5). An ANOVA is needed to conclude that this haplotype affects temperature sensitivity.

*Reviewer #3:*

The authors have generally met my major concerns and provide additional data as well as detailed explanations for the changes in the manuscript.

Remaining concerns:

1) The authors have clarified that the major concerns about the exclusion of accessions from the correlation of FLM expression and flowering time was largely based on a misunderstanding of the method and intention of the analysis. The changes to the text makes this much more clear now. The exclusion of the “exceptional" haplotypes raises the correlation from 0.13 to 0.46 in a linear regression analysis. Please also provide the corresponding p-values here to make sure that the p-values reflect a robust correlation effect (as is done in the further analysis of FLMß/FLC expression via multiple regression analysis). Please also describe how p-values were obtained for both cases then.

2) I appreciate the explanation regarding the use of the pooling strategy and the inclusion of statistics. However, no information on the performed test is given in the figure caption. was this also a t-test as mentioned in the methods? If so, please state whether a two-sided t-test and correction for multiple testing was implicated. Otherwise perform an ANOVA with suitable post-hoc test.

---

## [Author Response]

*[…] There are four key concerns that need to be addressed prior to any final decision.*

*1) The FLM expression to flowering correlation is not sufficient. The exclusion of multiple accessions was considered not justified. A multiple regression approach with FLC expression included in the model is a way that may account for this and would be a more correct statistical test of the hypothesis.*

Please see our reply to reviewer 2, comment 2 and Comment 3.

*2) There are other significant statistical concerns that may affect the core claims from the manuscript such as temperature sensitivity, etc. These must be corrected and the claims either supported or changed pending the results.*

Please see our reply to reviewer 2, comment 3 and reviewer 3 comment 2.

*3) The role of vernalization in FLM expression and variation thereof needs to be assessed to allow for the data to be properly integrated into the broader model. Given that the integration into the broader model is the critical interest in this paper.*

Please see our reply to reviewer 2 comment 3.

*4) There was concern about including FLMd if it is not found in the measurements? This should be addressed.*

See our reply to reviewer 2 comment 1.

*Essential revisions:*

*Please also address the following concerns that will improve the manuscript.*

*Reviewer #2:*

*The authors present a detailed search for the genetic basis of variation in ambient temperature sensitivity for flowering time in Arabidopsis. Focusing specifically on the control of the expression of FLM which has been previously linked to temperature sensitivity, the authors draw on a range of modern and innovative tools to pinpoint alleles conferring variation FLM expression within the species. They identify the major alleles controlling both basal expression and temperature sensitivity of FLM segregating at high frequency in the species and confirm effects of these alleles using transgenics. Overall, this study is important in its demonstration of how modern tools in Arabidopsis can be used to rapidly identify the specific base-pairs underlying natural variation in important traits. While I think that the major conclusions seem robust and well-supported, this is a very complicated paper and I do have concerns about specific analyses and interpretations.*

*1) My main concern is with the central focus on the comparison between two isoforms of FLM (FLMb and FLMd). A recent study showed that the qPCR assays targeting these isoforms actually pick up other isoforms, many of which produce truncated proteins (Sureshkumar et al. 2016 Nature Plants). Here, the authors note that they can't even find a true FLMd transcript (out of 163 tested). It is not clear what if any conclusions can be drawn from the FLMd assay, and so devoting so much space throughout the article and figures to FLMd is not useful and distracting. On the other hand, this and earlier studies the FLMb assay does seem to be useful for assaying FLM activity, so I am comfortable with these results.*

The major concept of our study had been to elucidate regulatory regions of FLM transcription and this also included analysis of FLMd. Rejecting the importance of FLMd was not a prerequisite but an important result of the study. It is true that the finding that FLMd may not have a function in flowering time control had already been a conclusion from our earlier work (Lutz et al., 2015). On the other side, the two papers that led to the original model, which now has to be considered invalid, were published very prominently (Nature, Science), reached a very large readership and are still frequently discussed and cited, e.g. in a very recent review (Deng and Cao, 2016, Current Opinion in Plant Biology). We therefore also felt that it may be used as an argument against our current data analysis and manuscript if we ignored FLMd.

We also felt that the two recent studies that put the model that attributed a functional role to FLMd into question (Lutz et al., 2015; Sureshkumar et al.,2016) did not yet convincingly proof that any regulatory role for FLMd could be completely rejected. We therefore included FLMd in the present study to test a putative regulatory role of FLMd with our new genetic materials and testing FLM expression in different environmental conditions. We had discussed internally and before submission of the original manuscript whether it was possible to reduce the data load by omitting FLMd data but came to the conclusion that this would weaken the significance of the data set. We have now reexamined this point and made the following changes to reduce the FLMd data load:

We have moved the FLMd data of Figure 5 to Figure 5—figure supplement 1. We removed the FLMd data from Figure 8—figure supplement 1 and the respective section in the text since, at this point, the conclusion that FLMd was not important had already been phrased.

That the primers used to detect FLMd may also amplify other minor isoforms was published comparatively late during our analysis (Sureshkumar et al., 2016). We agree that this issue needs to be addressed in the manuscript and discuss this point briefly in the Discussion:

“Our findings thus support more recent studies suggesting that the *FLM-δ* splice variant may be biologically irrelevant {Lutz, 2015 #118;Sureshkumar, 2016 #96}. One of these studies also identified a large number of biologically irrelevant transcripts that could be identified with the primer combination used here for the detection of *FLM-δ* {Sureshkumar, 2016 #96}. Our data thus support the conclusion that none of these amplification products has a biologically important function {Sureshkumar, 2016 #96}."

We have also modified Figure 1 and now show the position of the primers and specify in the graphs, which primer combinations had been used for the respective amplification product.

*2) Secondly, I am concerned with the results presented in Figure 8 – the correlations between FLM expression in accessions and a) expression in constructs and b) flowering times of the accessions. In order to find a correlation between FLM expression in accessions and in the transgenic lines with the corresponding constructs, the authors dropped an "outlier" point (1 each for FLMb and FLMd). No justification for this is given beyond the point not fitting expectations. It's not clear from Figure 8—figure supplement 1 that these points are really more outliers than any others. And, it's not that this is really a single point – on the y-axis this is the average expression of ~10 accessions, so why should they all be dropped?*

We apologize for having used the term “outlier” in an apparently misleading manner. In this analysis, we asked whether one of the nine haplotype groups influenced FLMb levels in a different manner from the others. We thereby examined whether it had a kind of outlying regulatory role. For this purpose, we performed the analysis by excluding, one by one, each of the nine haplotypes and calculated R^2^. None of these exclusions substantially changed R^[73]^ except when PRO2+^AAAAC^/INT6+^CAA^ was excluded. When this haplotype was excluded R^[73]^ increased from 0.13 to 0.46 indicating that this haplotype substantially and negatively influenced our analysis. Herein, our aim had not been to force the linear model to fit our expectations but to identify a PRO2+/INT6+ haplotype, which possesses a differential effect when tested in transgenic lines when compared to accessions. We have now clarified the legend of Figure 8, removed the term outlier, and improved the description of the findings in the text. We also removed the correlation analysis of FLMd from Figure 8—figure supplement 1.

Legend 8A:

**“**Figure 8. *FLM-ß* expression shows high correlation with flowering time in *Arabidopsis* accessions. […] Data of the transgenic lines was taken from Figure 4. The grey area indicates the 95% confidence interval.”

Results:

**“**PRO2+ and INT6+ polymorphisms contribute to global variation of *FLM* levels: The nine PRO2+ and INT6+ haplotypes tested in transgenic experiments were present in 579 (69%) of all 840 accession with available genome sequence information (Figure 8—figure supplement 1). […] This indicated that PRO2+^AAAAC^ INT6+^CAA^ has an exceptional effect on *FLM-ß* in the accessions than cannot be predicted based on the results from the transgenics (Figure 4, Figure 8 and Figure 8—figure supplement 1).”

*3) For the flowering time comparison, the lines were vernalized. However, FLM is known to be repressed by vernalization (as the authors note), so how could it be affecting flowering, and might there be variation among lines in how FLM responds to the vernalization? It seems more likely that other loci that are correlated with FLM are controlling flowering time here. The authors should either measure FLM expression in these vernalized plants and correlate that with flowering, or repeat the experiment without vernalization where FLM and FLC expression in these lines has already been measured. A better way to control for FLC variation would be to statistically account for its effects with multiple regression.*

We have extracted from the literature that the role of vernalization on FLM expression is rather unclear. Ratcliffe et al. (2001, Plant Phys.) observed decreased *FLM* levels after six to eight weeks of vernalization, even though *FLM* seems not to be as strongly repressed by vernalization as *FLC*. Scortecci et al. (2001, Plant Journal) did not find an effect on *FLM* transcript abundance after 30 days of vernalization. However, this observation was revisited in Sung et al. (2006, Genes and Development), who showed that *FLM* is repressed by vernalization through epigenetic mechanisms. Finally, Lee et al. (2013, Science) did not observe any effect of vernalization on *FLM* abundance whereby no information on the duration of the vernalization treatment was provided. Even though a repressive effect of vernalization on *FLM* abundance seems to be the consensus we suppose that the effect of vernalization on *FLM* abundance needs much more detailed investigation. Especially the duration of vernalization may be a critical factor. We had summarized this literature and findings in our original manuscript. Since we now present an improved analysis based on improved experimental design as described below, we have deleted the respective sections since they were no longer relevant.

Regarding the improved experiment as suggested by the reviewer, we considered it not insightful to conduct an experiment with vernalized plants (and including winter annuals) since the FLC-dependent vernalization pathway is epistatic to the effects of FLM.

Instead, we have now performed an experiment with summer annuals not requiring vernalization (all carrying PRO2+/INT6+ haplotypes of the described range). As many summer-annual accessions carry weak *FLC* alleles with residual low *FLC* abundance, which still are able to contribute to repression of flowering time {Duncan, 2015 #156;Coustham, 2012 #30;Li, 2014 #111}, we measured *FLM* and *FLC* transcript levels and tested their combined and single contribution to flowering time (rosette leaf numbers). Flowering time data was generated in a new experiment. Interestingly, we found a comparable significant contribution of FLMb to flowering time than in the first experiment (R^2^ = 0.21, before R^2^= 0.15), with a smaller, however not significant, contribution of the residual FLC levels (R^2^ = 0.119). We integrated the findings of this new experiment into Figure 8, changed Figure 8—figure supplement 2, changed the Results, Discussion, and figure legends.

Results:

“To examine the correlation between *FLM-ß* expression and flowering time, we determined flowering time of accessions at 15°C. […] Taken together, we concluded that PRO2+ and INT6+ alleles contribute to variations in *FLM-ß* levels and that *FLM-ß* accounts for flowering time in cool ambient spring temperatures in a diverse population of summer-annual *Arabidopsis* accessions (Figure 9).”

Discussion:

“The nine PRO2+/INT6+ haplotype combinations included in our transgenic experiments represent 69% of the world-wide PRO2+/INT6+ variation (Figure 8—figure supplement 1). […] Bearing these possible genetic and environmental interferences in mind, we regard the detected effect of *FLM-ß* on flowering (21%) in cool (15°C) temperatures as considerable and suggest that it is rather an underestimation.”

Figure legends:

**“**Figure 8. *FLM-ß* expression shows high correlation with flowering time in *Arabidopsis* accessions. […] (B) Correlation of *FLM-ß* transcript levels with flowering time (n = 8 – 10) as measured in rosette leaf number of summer-annual accessions. p = 0.011.”

**“**Figure 8—figure supplement 2: Geographic distribution of summer-annual accessions. (A) Geographic distribution of the 27 summer-annual accessions as described in Figure 8. (B) Correlation of *FLC* transcript levels with flowering time data (n = 8 – 10) as measured in rosette leaf number of summer-annual accessions. p > 0.05.”

*4) Finally, on the statistical side, the conclusions about "temperature insensitivity" for INT6+ alleles are not backed by statistics – the authors should explicitly test for a change in the temperature sensitivity between FLM-Col0 and INT6+. From the graphs, it appears temperature sensitivity is reduced, but certainly not eliminated (Figure 6).*

To address this comment, we have re-evaluated the observation with a new measurement and corrected the calculation and integrated statistical tests. The results and the conclusions were found to be consistent with those presented in the initial manuscript.

*Reviewer #3:*

*In “Natural haplotypes of FLM non-coding sequences fine-tune flowering time in ambient spring temperatures in Arabidopsis" Lutz et al. address the biological relevance of FLM splice variants for variation in flowering time in natural accessions and identify non-coding sequence polymorphisms that contribute to the regulation of basal expression levels and the differential, thermo-responsive generation of the β and δ splice variants. As such, this work provides novel insights into the regulation of flowering in the ambient temperature range and manages to dissect polymorphisms which are relevant for basal FLM expression and temperature-induced effects on the β and δ splice variants.*

*The manuscript is overall very well written and the authors provide extensive data which has apparently been subjected to comprehensive and sound statistical analyses. The data is presented in an attractive manner. Main manuscript figures are supported by extensive supplementary material including the presentation of extensive control data, which is greatly appreciated. In some cases the figure descriptions or legends should be extended. Due to the vast amount of data condensed in the figures, the descriptions (at least in the text) should be detailed enough for a wide readership to follow the authors through their argumentation.*

*1) The authors state that “The presence of intron 1 is sufficient to restore the basal expression of FLM” – while it does restore expression levels to detectable levels, the expression level of the β splice variant at 15°C is considerably lower than the WT, whereas the FLMδ levels are similar the WT at this temperature. This would indicate that intron 1 alone cannot fully account for basal levels of FLM β.*

The reviewer is correct in stating that Figure 1 may invite the conclusion that intron 1 does not fully account for basal FLMb levels. However, as we used transgenic lines for this analysis we suggest that no conclusions about the basal expression levels of native FLMb nor FLMd can be drawn from this analysis. Nevertheless, the relative change of transcript levels in response to changing temperature can be detected and compared to the WT construct, which is the main aim of this analysis. To avoid any misunderstandings, we normalized the transcript levels measured at 21°C to “1” and provided an improved description in the legend.

Figure legend:

“Figure 1. *FLM* intronic sequences determine basal and temperature-dependent *FLM* expression. […] For normalization, the respective values measured at 21°C were set to 1.”

*2) The use of pools of independent segregating T2 lines for transcriptional analysis seems unusual. While the controls presented in Figure 2—figure supplement 2 suggest that this can recapitulate the “natural” situation (why not subjected to statistical analysis?), it seems nonetheless risky to deliberately tolerate the presence of non-transgenics in the mix. Could the authors state their reasoning for this unusual procedure and provide the missing statistics here? Also, the phrase “replicate pools compromising…” should probably read “comprising”.*

By adopting this pooling strategy, we have followed procedures that were described in publications from leading flowering time labs (e.g. Coustham, 2012, Science; Caroline Dean lab). This is an attractive way to reduce the variability of transgene expression that result from positional effects in different transgenic lines. In view of the ample analysis of transgenic lines presented in our manuscript, this presented the most efficient way for reducing cost and time while still producing a meaningful data set. If we had used T3 generation plants, we would have had to propagate and test the segregation of over 3000 individual plants. We have modified (and thereby improved) the above-described strategy by pooling plants into four pools, which we designate replicate pools. We have also verified the trustworthiness of this pooling system by demonstrating that it is possible to precisely recapitulate the effects observed with the Col-0 and Kil-0 *FLM* alleles (Figure 2—figure supplement 2). The statistical analysis has now been integrated into the figure as requested by the reviewer.

It should also be noted that we verified that the lines are single insertion lines by scoring their segregation, which was possible due to the use of the FAST vector system, which allows the identification of transgenes by seed fluorescence (Shimada, 2010). Further, we compared the transcript levels to a Col-0 allele reference transgenic construct, which was obtained by the exact same procedure of pooling etc.

"Compromising" has been corrected to "comprising".

*3) I had a hard time to comprehensively grasp the information provided in Figure 3, especially when it comes to the color code and its use in Figure 3.*

Yes, it is true that the use of the same color in both panels may imply that the two data sets are related to each other, but they are not. We have now changed the colors in 3B to avoid this confusion.

*Figure 3—figure supplement 1 provides additional information, but still the color coding of haplotypes which is provided for three different regions (full length, intron 1 or intron 6) is confusing as the composition of the different haplotypes varies depending on the selected gene region (Figure 3—figure supplement 1). The authors may want to better explain these figures to enable a wide readership to follow their analysis and key points here.*

The reviewer is correct in stating that using similar colors for different haplotype regions may be confusing. Here, we used the same color code for each grouping by using the same color for the respective largest group of each the three comparisons. To clarify this, we have now adjusted the shade of these color codes and provided a better description.

Figure legend:

Figure 3—figure supplement 1: Haplotype analysis based on 45 SNPs from 776 accessions. […] The color coding for haplotype group numbers, which illustrates the group number and group size as shown in (D) and (E) is specified below the graph.”

*Also, in the caption of Figure 3—figure supplement 1 the haplotype “Numbers on the left” probably refer to (D) not (E)?*

Yes, thank you. We have corrected this.

*4) Figure 5: Please specify again in the figure caption that this data corresponds to 40 selected accessions. Why was the FLM β level not determined in the vernalized plants? Even though FLC effects seem to be generally negligible, it would have been as easy to do the analysis on the same material.*

We have improved the experimental design and have now included an improved data set by using summer annual accessions with minor FLC activity. Such an experiment had been suggested by reviewer #2. Please also see our reply to reviewer 2, comment 3.

*5) In Figure 5—figure supplement 1: removing the lower quartile (or more!) of flowering time data for the correlation analysis by simple assumption that these represent the non-transformed Nd-1 individuals of the segregating population is questionable. While this may be an attractive assumption, it negates the fact that each line pool may represent its individual set of variances (also present in ND-1 itself). This would at least require a spot check for the presence of transgenes in this pool or the percentage of “clean” Nd-1 individuals. In principle, a simple test by PCR would have provided a solid answer to that. As is, the authors should include these lines in the correlation analysis as the factual genotype is not known.*

We have provided the full data set without corrections (Figure 5—figure supplement 1/E) as well as the corrected data set (Figure 5). For the data presented in Figure 5, we precisely removed the lower quartile and emphasize that we had confirmed that all lines carried single locus inserinots, using a fluorescence marker. Thus, 25% of the plants should be wild type segregants from the transgenic lines and these should correspond to the early flowering individuals since the genetic background for the transgenic analysis had been the *FLM* deletion accession Nd-1. In this analysis, the homozygous wild type lines Nd-1 and Col-0 are not included since they do not segregate and the correction procedure of removing the lower quartile could not be applied.

Regardless of this correction, very similar R^2^ values were obtained with the corrected (R^2^ = 0.94) and uncorrected data (R^2^ = 0.89). We assume that the high correlation between FLMb levels and flowering time is generally very high, regardless of the only small improvements in the correlation analysis when using the corrected or uncorrected flowering time values.

We have clarified the text and added further detailed information.

Figure legend:

“Figure 5. *FLM-ß* expression shows high correlation with flowering time of transgenic plants. […] An analysis using the uncorrected data is shown in Figure 5—figure supplement 1.

[Editors' note: further revisions were requested prior to acceptance, as described below.]

*The manuscript has been improved but there are some remaining issues that need to be addressed before acceptance, as outlined below:*

*There are still some key issues on interpretation and analysis that need to be addressed. None of these require new wet lab experiments and are relatively easy and quick to clean up. All felt that this is necessary.*

*Reviewer #2:*

*I thank the authors for their improvements to the paper – the figures especially are more clear now. I still have the following concerns about the overall conclusions and analysis. These relate to three important conclusions:*

1) That FLM variation contributes to flowering time variation across a diverse panel of summer annual accessions.

*The new experiment to test this is very helpful, and does provide good evidence that FLMB expression level is important for flowering time variation. My concern is with interpretation: The authors provide several explanations for why the correlation between flowering and FLMB expression in this accession set is lower than in the transgenics. This is trivial because there is much more variation within each haplotype class in the accessions than the transgenics, because they vary at many loci across the genome. I think the real question should be: Is the slope of the relationship between FLMB and flowering time significantly less in the accessions? This would ask if the equivalent change in FLMB expression would cause the same change in flowering. This is a more meaningful metric than change in R2. The correct test would be to include both accessions and transgenics in the same model (flowering_time ~ FLMB * genotype_class), and ask if the interaction is significant.*

We performed the suggested test for interaction FLMB:genotype_class and found that the slopes differ (p = 0.0321). We included this analysis in the results (subsection “PRO2+ and INT6+ polymorphisms contribute to global variation of FLM levels”, last paragraph) and Discussion.

2) That the 9 promoter haplotypes explain the variation in FLMB expression in the accessions.

*This is the aim of the analysis in Figure 8 and Figure 8—figure supplement 1/E. The main claim is that the expression variation induced by these haplotypes in the transgenics is correlated with the expression variation of accessions carrying the same haplotypes. This conclusion only holds when one of the 9 haplotypes is excluded. The explanation given for excluding this haplotype is that the correlation goes up after removing this. I did a quick simulation study and found that using this algorithm you'd expect that the best correlation found by dropping each of the 9 pairs would be greater than 46% about 21% of the time even if there were actually no correlation at all. So, this is really not very strong evidence that the haplotype effects are similar. Certainly, the statement "We found that the PRO2+/INT6+ effects, as detected in transgenic experiments (R2 = 0.94), partially explain the FLM-ß expression variation in natural Arabidopsis accessions (R2 = 0.46)" is not supported, because this 46% number is only true when you exclude those accessions that don't fit.*

*As above, I think a more appropriate analysis would be to ask if the sizes of the differences between the haplotypes is the same in the transgenics and the accessions (i.e., is the slope of the graph different from 1?). A more straightforward answer to the question of how much variation in FLMB is explained by these haplotypes is simply to report the R2 for the analysis shown in Figure 8—figure supplement 1.*

We understand the reviewer's concern and deleted the analysis excluding the exceptional haplotype from Figure 8 and reported the R2 of the full dataset shown in Figure 8—figure supplement 1 (R^2^ = 0.13) as suggested. We changed the respective parts of the text, worded the related conclusions less strongly (subsection “PRO2+ and INT6+ polymorphisms contribute to global variation of FLM levels” and Discussion) and changed Figure 8 accordingly.

3) That the INT6+(CAA) haplotype controls temperature sensitivity.

*This conclusion is important because of the model that FLM controls temperature sensitivity of flowering. New data are presented here relative to the previous version, though where they came from is not clear. The correct test for a change in temperature sensitivity is to ask if the interaction between genotype and temperature is significant (expression ~ Genotype + Temp + Genotype:Temp). The n.s. effect of temperature on expression for INT6+ is not evidence of no temperature effect, only a lack of evidence for a temperature effect. The figure caption states the statistics are done based on a t-test. A t-test can't be used to test for an interaction (except in 6C/D if the 5 transgenic lines per genotype were used as the replicates, n = 5). An ANOVA is needed to conclude that this haplotype affects temperature sensitivity.*

The new data was generated by measuring FLM-β levels with qRT-PCR using the same samples. As suggested by the reviewer, we performed an interaction test between genotype and temperature of the data shown in Figure 6. We found a significant change in temperature sensitivity (p = 0.012259). We integrated this test in the Results section subsection “The INT6+CAA polymorphism confers temperature-insensitive FLM expression and flowering”) and the figure legend. As the change in temperature sensitivity was now reported by testing the interaction of genotype to temperature, we removed the t-tests from Figure 6 and B.

*Reviewer #3:*

*The authors have generally met my major concerns and provide additional data as well as detailed explanations for the changes in the manuscript.*

*Remaining concerns:*

*1) The authors have clarified that the major concerns about the exclusion of accessions from the correlation of FLM expression and flowering time was largely based on a misunderstanding of the method and intention of the analysis. The changes to the text makes this much more clear now. The exclusion of the “exceptional" haplotypes raises the correlation from 0.13 to 0.46 in a linear regression analysis. Please also provide the corresponding p-values here to make sure that the p-values reflect a robust correlation effect (as is done in the further analysis of FLMß/FLC expression via multiple regression analysis). Please also describe how p-values were obtained for both cases then.*

Please see our reply to comment #2 of reviewer #2. According to his comment and reasoning, removing the exceptional haplotype may not be appropriate. Therefore, we decided to simply report the R2 shown in Figure 8—figure supplement 1 in the text. For further details please see our answer to comment #2 of reviewer #2.

*2) I appreciate the explanation regarding the use of the pooling strategy and the inclusion of statistics. However, no information on the performed test is given in the figure caption. was this also a t-test as mentioned in the methods? If so, please state whether a two-sided t-test and correction for multiple testing was implicated. Otherwise perform an ANOVA with suitable post-hoc test.*

A t-test was applied previously. We now performed an ANOVA and Tukey HSD post-hoc test. The interpretation remained unchanged compared to the previously applied t-tests. We changed Figure 2—figure supplement 2 and the respective figure legend accordingly.